# Emerging many-to-one weighted mapping in hippocampus-amygdala network underlies memory formation

Jun Liu, Arron F. Hall ● & Dong V. Wang ● ✉

Memories are crucial for daily life, yet the network-level organizing principles governing neural representations of experiences remain unknown. Employing dual-site in vivo recording in freely behaving male mice, here we show that hippocampal dorsal CA1 (dCA1) and basolateral amygdala (BLA) utilize distinct coding strategies for novel experiences. A small assembly of BLA neurons emerged active during memory acquisition and persisted through consolidation, whereas most dCA1 neurons were engaged in both processes. Machine learning decoding revealed that dCA1 population spikes predicted BLA assembly firing rate, suggesting that most dCA1 neurons concurrently index an episodic event by rapidly establishing weighted communication with a specific BLA assembly – a process we term "many-to-one weighted mapping." We also found that dCA1 reactivations preceded BLA assembly activity preferably during elongated and enlarged dCA1 ripples. Using a closed-loop strategy, we demonstrated that suppressing BLA activity after large dCA1 ripples impaired memory. These findings highlight a many-to-one weighted mapping mechanism underlying both the acquisition and consolidation of new memories.

Converging evidence suggests that the formation of new memories involves a complex neural network of brain regions, including the hippocampal dorsal CA1 (dCA1) and basolateral amygdala (BLA). How neuronal populations within these brain regions encode and exchange information is yet to be fully understood[1–4]. The formation of episodic memories, or memories of events, can be divided into two processes: memory acquisition and consolidation. Memory acquisition is rapid, indexing episodic events almost instantaneously; however, these memories are often fragile and only become stabilized after a slow transformation process known as memory consolidation[5,6]. This process critically involves sleep, with dCA1 ripples, fast oscillations (100–300 Hz) that occur predominantly during slow-wave sleep and immobility, playing a crucial role[5–7]. Despite this understanding, it is still unclear how the dCA1–BLA network enables such rapid indexing and gradual consolidation of new memories.

Memories of events consist of three primary components: where, when, and what[8,9]. Previous studies have demonstrated that the dCA1 encodes where and when by sequences, such that a substantial portion of dCA1 neurons fire sequentially during spatial navigation or performing time-based tasks[10,11]. In contrast, no consensus has been reached regarding how dCA1 neurons encode specific events (what). Given the continuous unfolding of episodic events in our daily life, the encoding of new events becomes intricately intertwined with the recall of past experiences. Consequently, dCA1 neurons become highly entangled in the process of encoding both ongoing and past event information[12], making it challenging to disentangle dCA1 spike patterns that specifically encode individual events.

Presently, there are two major hypotheses regarding how the hippocampus encodes episodic events (what). One prominent view holds that "(event) information is sparsely encoded in distributed ensembles of hippocampal neurons"[13]. However, there is limited evidence supporting this sparse-coding hypothesis by dCA1 neurons[13–15]. In fact, recent findings suggest the contrary, revealing that a substantial portion of dCA1 neurons (up to 50%), instead of a minority, are

Department of Neurobiology & Anatomy, Drexel University College of Medicine, Philadelphia, PA 19129, USA. ✉e-mail: dw657@drexel.edu

activated during fear memory procedures[16–19]. Moreover, ~30–42% of dCA1 neurons exhibit co-activation when exposed to two different contexts[16,20], indicating a substantial overlap of dCA1 neurons in encoding distinct contextual memories. These findings generally contradict the sparse-coding hypothesis, at least within the dCA1 region.

The other influential view suggests that the hippocampus functions to link or bind, rather than encode event information that is otherwise represented in a distributed neural network[21,22]. This viewpoint, however, fails to address the precise mechanisms through which dCA1 activity effectively achieves such linking/binding, and it has not been directly tested experimentally. To address this knowledge gap, our study employs a different approach to deduce dCA1 encoding

principles by computing activity correlations between dCA1 neurons and specific BLA assemblies. This approach leverages recent findings that BLA neurons form distinct assemblies to represent salient events and memories[23–25].

## Results

### Emerging dCA1–BLA communication underlies memory formation

To investigate the encoding principles and communication dynamics of the dCA1 and BLA neuronal populations, we conducted dual-site in vivo recording of the dCA1 and BLA (up to 16 tetrodes per site; Fig. 1a). All mice received a contextual fear memory procedure that consisted of pre-training sleep, training, post-training sleep, and fear

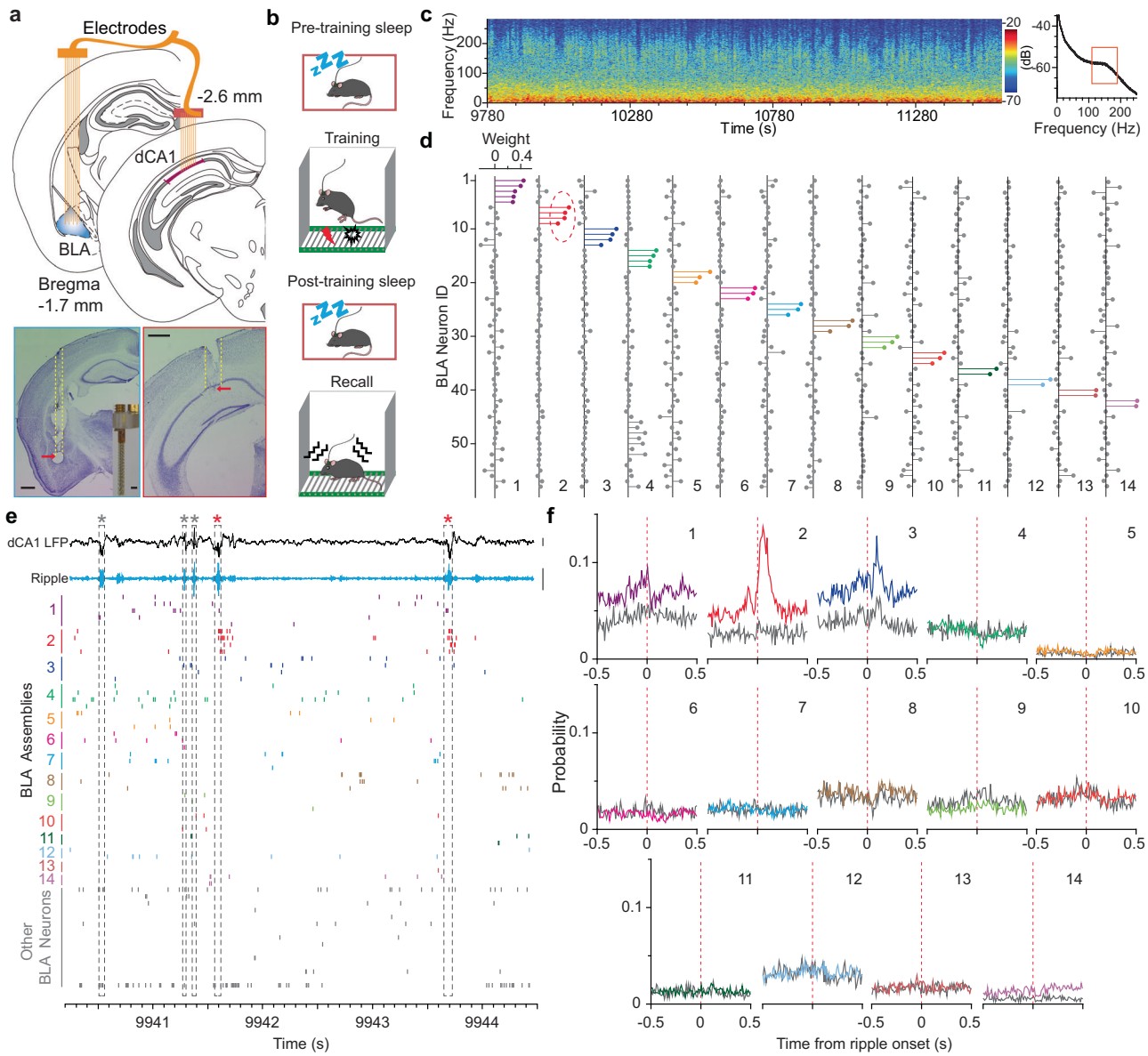

**Fig. 1 | Emerging dCA1 ripple-to-BLA assembly communication underlying memory consolidation. a** Schematic illustrating dual-site recording (top) and representative brain sections showing recording sites in the BLA and dCA1 (bottom). Inset, a self-constructed 16-tetrode array[61]. Scale bars, 0.5 mm. **b** Schematic of the contextual fear memory procedure. **c** Spectrogram (left) and power spectral density analysis (right) of representative dCA1 LFP recorded during a post-training sleep session. **d** Independent component analysis identified 14 assemblies based on

the spikes of 58 BLA neurons recorded during the same sleep session as shown in panel c. **e** Representative dCA1 LFP, band-pass filtered ripples (100–250 Hz), and spikes of the same 58 BLA neurons as shown in panel d. Note that assembly 2 exhibits robust activation after a subset of dCA1 ripples (red stars). Scale bars, 0.5 mV. **f**, Cross-correlograms between dCA1 ripples (n = 2147) and the 14 BLA assemblies. Grey and color lines indicate pre- and post-training sleep, respectively.

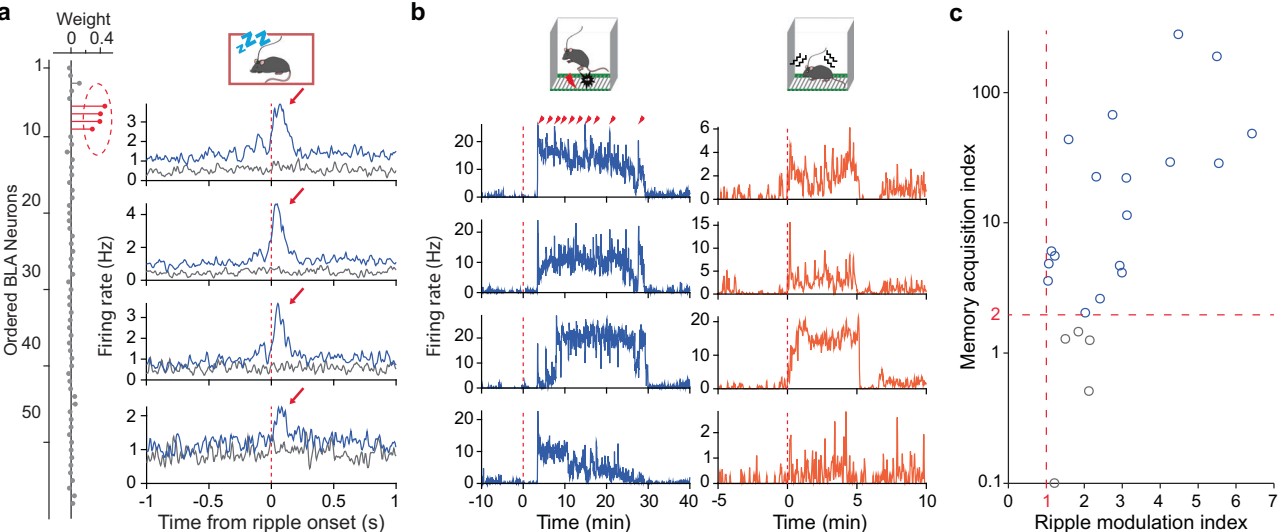

**Fig. 2 | A unique assembly of BLA neurons underlying memory formation.**
**a** Left, one ICA-identified BLA assembly (assembly 2 as shown in Fig. 1d). Right, cross-correlograms between individual neurons of the BLA assembly and dCA1 ripple events during pre-training (grey lines) or post-training sleep (blue lines). **b** Rate histograms of the same four BLA neurons during fear memory acquisition (left; 10 shocks) and retrieval (right). Time 0 indicates the placement of mice in the shock chamber. Note the little change in activity between −10−3 min before shock administration during acquisition. **c** BLA memory neurons (blue circles) are defined as neurons with a ripple modulation index (RMI) > 1 (neurons with RMI < 1 are not shown) and a memory acquisition index (MAI) > 2. RMI = (Peak$^{post}$ − Peak$^{pre}$) / Baseline. Peak$^{pre}$ and Peak$^{post}$ are peak firing rates of BLA neurons within 150 ms after ripple onsets (arrows in a) during pre- and post-training sleep, respectively; Baseline is the mean firing rate during SWS. MAI = Mean$^{post}$ / Mean$^{pre}$. Mean$^{pre}$ and Mean$^{post}$ are mean firing rates of BLA neurons before (0−3 min) and after the first shock administration (3−30 min), respectively.

recall test (Fig. 1b). Neuronal spikes and local field potentials (LFPs) were recorded throughout the entire process, which enabled us to study dCA1–BLA neuronal ensemble dynamics at each memory stage, including acquisition, consolidation, and retrieval.

We first identified slow-wave sleep (SWS) stages based on prominent dCA1 delta[26] and ripple oscillations (Fig. 1c). Next, we conducted an independent component analysis (ICA) of population spikes recorded during SWS to identify major BLA assemblies, i.e., small groups of co-activated neurons[27]. As an example, the ICA identified 14 assemblies based on the activity of 58 BLA neurons recorded during post-training SWS (Fig. 1d; Supplementary Fig. 1). We observed that one selective BLA assembly (#2) robustly increased its activity after a subset of dCA1 ripples (Fig. 1e). Subsequent cross-correlation analysis confirmed this observation and revealed additional BLA assemblies that showed decreased activity (# 1&8), decreased activity followed by a rebound (#3), or little change of activity (Fig. 1f).

On average, the ICA algorithm identified 8.8 ± 1.0 BLA assemblies (mean ± s.e.m.; 2–6 neurons per assembly) across datasets recorded from 10 mice, with a minority of BLA neurons not forming assemblies (Fig. 1d; Supplementary Figs. 2–4). Notably, often one BLA assembly (or neuron) from each dataset showed robust dCA1 ripple-modulated activation during post-, but not pre-training sleep (Fig. 1f; Supplementary Figs. 2–4), suggesting the emergence of dCA1 ripple-to-BLA assembly communication after learning. Although there are other ripple-modulated BLA assemblies, they often exhibit decreased activity immediately after ripple events, or little modulation between pre- and post-training sleep (Fig. 1f; Supplementary Figs. 2–4). Based on a permutation test, the proportions of ripple-modulated BLA assemblies are significantly higher than chance in each of the datasets during post-training sleep, as well as in most recording sessions (7/10) during pre-training sleep (P < 0.05; see Methods).

To provide further evidence that these ripple-modulated BLA assemblies (or neurons) are involved in memory processes, we analyzed their activity during memory acquisition and memory retrieval. Our results revealed that most of the post-training ripple-modulated BLA neurons (~78%; 18/23) exhibited robust prolonged activation

during memory acquisition and retrieval stages (Fig. 2; Supplementary Figs. 2&3). These results suggest the involvement of a unique BLA assembly engaging in multiple memory stages, including acquisition, consolidation, and retrieval.

Subsequently, we termed a BLA assembly as a "BLA memory assembly" if it displayed robust activation during memory acquisition and ripple-modulated activation during post-, but not pre-training sleep (Fig. 2c; Supplementary Fig. 4). Individual neurons within each BLA memory assembly or single BLA neurons that met the above criteria were likewise termed "BLA memory neurons." Notably, only one BLA assembly (or neuron) was identified as a memory assembly (or neuron) in each dataset based on the above criteria. Overall, these BLA memory neurons comprise ~5.3% (18/341) of the BLA neuronal population, indicating a sparse coding feature. Additionally, we found that communications between dCA1 ripples and BLA memory assemblies (or neurons) were more prevalent during early-stage sleep compared to later-stage sleep after the contextual fear training (Supplementary Fig. 5).

We next investigated whether BLA assemblies preexisted prior to learning or were newly formed during the learning process. Our ICA assembly analyses, based on separate pre-training or post-training sleep datasets, largely identified the same groups of BLA neurons as assemblies (Supplementary Fig. 1). Importantly, further analyses revealed that most BLA assemblies identified based on post-training sleep were already highly active during pre-training sleep (Supplementary Fig. 1), despite that their activity did not necessarily coincide with dCA1 ripple events during pre-training sleep. These findings suggest that memory formation primarily recruits preconfigured or preexisting BLA assemblies, rather than forming new assemblies, a notion consistent with recent findings[3].

## Memory-associated ripples are enlarged and elongated

We next asked if the dCA1 ripple-to-BLA assembly communication conveyed unique information. To address this, we classified dCA1 ripples into two categories: memory and non-memory associated ripples. Memory-associated ripples were defined if they occurred

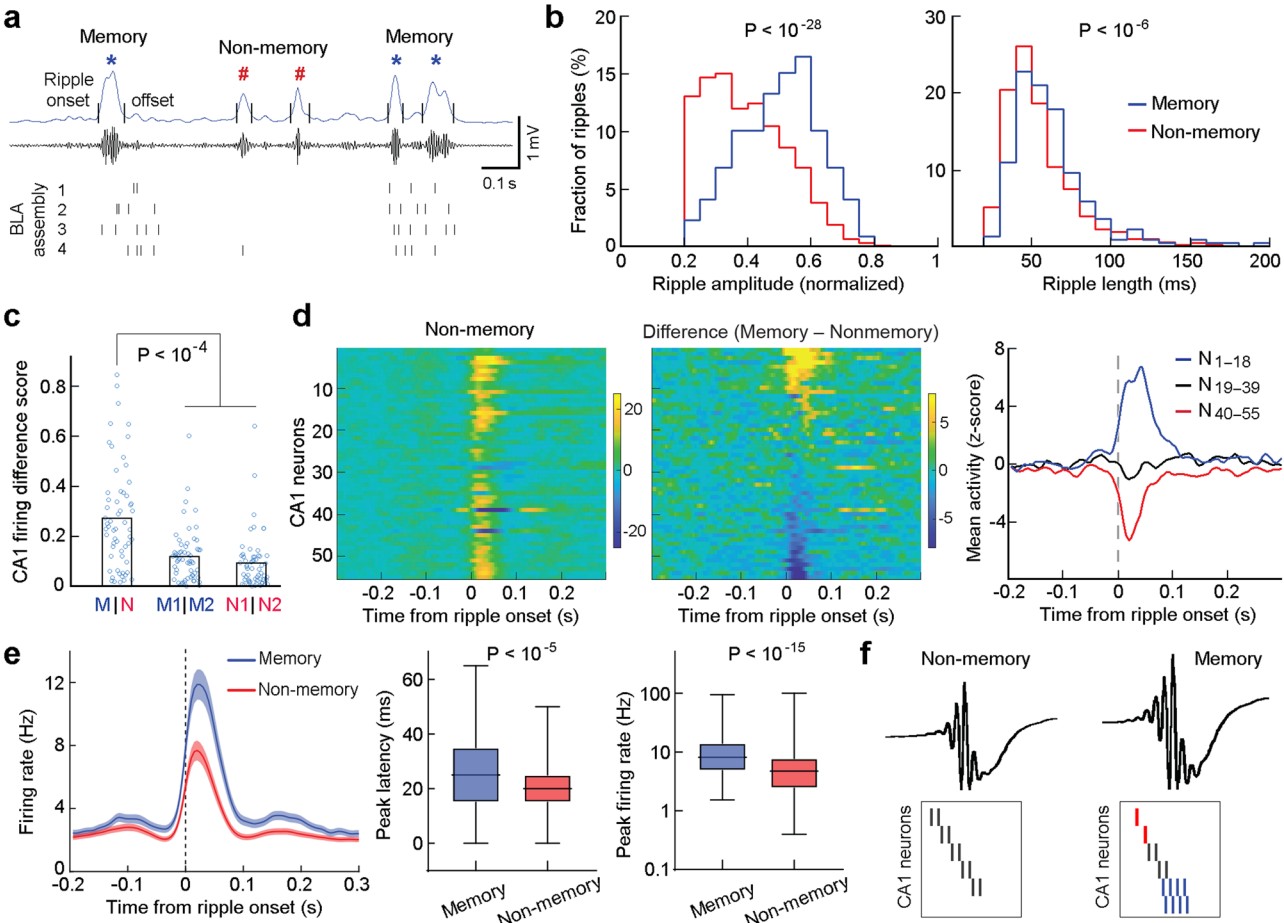

**Fig. 3 | Memory-associated ripples are enlarged and elongated. a** The BLA assembly (assembly 2 as shown in Fig. 1 d/e) exhibits activation at larger/longer ripples. Overall, 219 memory and 2390 non-memory ripples were classified in this post-training recording session (~1 h). **b** Memory-associated ripples have significantly larger amplitude (left) and longer duration (right). **c** Memory-associated ripples convey distinct contents. M/M1/M2 and N/N1/N2 indicate memory and non-memory ripples, respectively (see Methods). **d** Left, z-scored activity of dCA1 neurons (n = 55; recorded simultaneously with the 58 BLA neurons as shown in Fig. 1 **d/e** in relation to the onset of non-memory ripples. Middle, activity difference of the same 55 dCA1 neurons in relation to memory vs. non-memory ripples. The neurons are arranged in the same order in the two heatmaps. Right, mean activity of the upregulated (#1–18), unmodulated (#19–39), and downregulated dCA1 neurons (#40–55). **e** Left, Mean activity ( ± s.e.m.) of all dCA1 neurons that were activated during memory or non-memory ripples (n = 234 vs. 365 neurons, from 10 mice). Middle & Right, dCA1 neuronal activity was higher and peaked later at memory-associated ripples. In box plots, the central lines indicate the median value, whereas the box edges and whiskers mark the interquartile ranges and limits, respectively. ****$P < 0.0001$. **f** A proposed model of large/elongated ripples that contain distinct contents. Red ticks denote downregulated activity, blue ticks denote upregulated or emerged activity, and black ticks denote little change in activity of dCA1 neurons. All statistics are Wilcoxon two-sided rank-sum test.

coinciding with BLA memory assembly activation, while the remaining ripples were defined as non-memory ripples (Fig. 3a; see Methods). Our analyses revealed that memory-associated ripples had significantly larger amplitude and longer duration (Fig. 3b; Supplementary Figs. 2&3). Moreover, these memory-associated ripples contained distinct contents, i.e., spikes of different groups of dCA1 neurons (Fig. 3c). Comparing memory-associated ripples to non-memory ones, most dCA1 neurons showed further firing changes on top of their already increased activity: about one third showed upregulated activity while another one third showed downregulated activity (Fig. 3d; Supplementary Figs. 2&3). These findings suggest that a unique population activity pattern of dCA1 neurons communicates with a selective BLA assembly for relevant memory consolidation.

We observed that the increased dCA1 activity during memory-associated ripples tend to be prolonged (Fig. 3d, right). To verify this, we conducted an unbiased analysis on all dCA1 neurons that increased activity during memory vs. non-memory ripples. Our analyses confirmed that dCA1 neuronal firings were elongated and more robust during memory-associated ripples across animals (Fig. 3e; Supplementary Figs. 2&3). These elongated dCA1 ripples and enhanced firings

likely signify the consolidation of newly formed memories[28]. Taken together, we propose a model of ripple-associated memory consolidation, as depicted in Fig. 3f. During non-memory ripples, most dCA1 neurons exhibit a baseline probability of activation that represents preconfigured rigid motifs[29,30]. Upon memory consolidation, relevant dCA1 neurons exhibit upregulated and/or prolonged activity during ripples, while the remaining dCA1 neurons show downregulated or no change in activity, which collectively represent specific information coding.

### dCA1 population spikes predict BLA memory assembly firing rate

We next asked if dCA1 population spikes can predict the firing rate of BLA memory assemblies or neurons. To address this, we implemented generalized linear model (GLM) machine learning decoding[31]. Given that dCA1 neuronal activity preceded BLA memory assembly activity by ~35 ms (Fig. 4 a&b), we used a shift of 35 ms between dCA1 and BLA neurons for the GLM decoding. Overall, dCA1 and BLA neuronal spikes across multiple sliding windows (lasting 100 ms; Fig. 4c) before or after dCA1 ripples were extracted: 50% of them were used for training of the

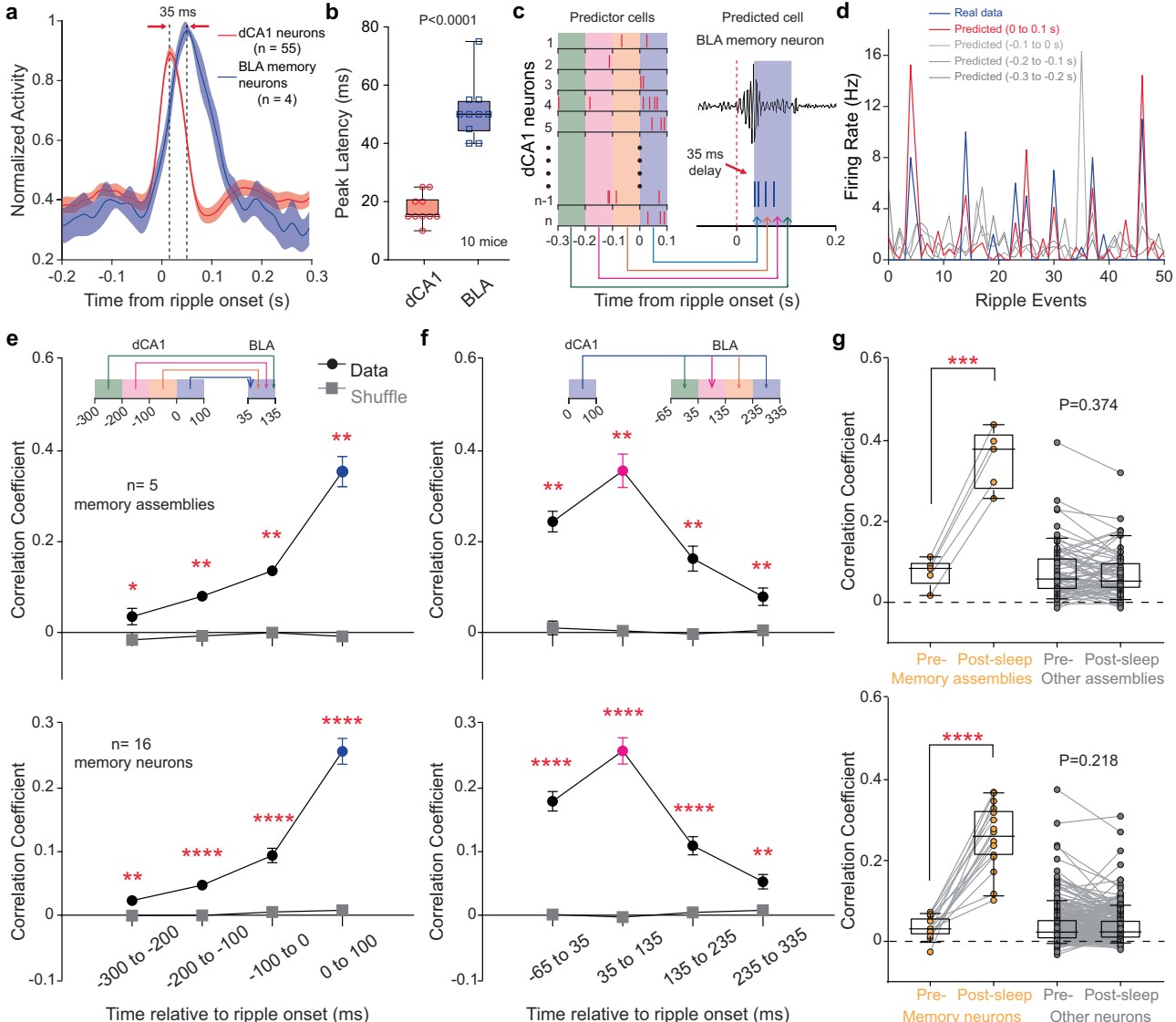

**Fig. 4 | dCA1 population spikes can decode BLA assembly firing rate. a** Mean peri-ripple histograms (± s.e.m.) of simultaneously recorded dCA1 neurons and one BLA memory assembly. **b**, Peak firing rate latencies of dCA1 neurons and BLA memory neurons. *N* = 10 mice, *P* < 0.0001, two-tailed paired *t* test. **c**, Diagram of GLM decoding. Spikes of dCA1 and BLA neurons were binned at multiple 100-ms time windows. "Predictor cells" are dCA1 neurons; "Predicted cell" is a BLA assembly or neuron. **d** Decoding the firing rate of one representative BLA memory assembly based on population dCA1 spike counts sampled at multiple bins as shown in c. **e, f** Top panels, correlation coefficients between real and predicted firing rates of BLA memory assemblies across multiple bins. Note that only recordings of >20 dCA1 neurons simultaneously were used for the decoding. Wilcoxon two-sided rank-sum tests revealed significant differences between the real and shuffled data across all time windows. Data are presented as mean values ± s.e.m., *n* = 5 memory assemblies, **P* < 0.05, ***P* < 0.01; ****P* < 0.001, *****P* < 0.0001. Bottom panels, similar to top panels except showing BLA memory neurons (*n* = 16 memory neurons). **g**, Top panels, correlation coefficients between real and predicted firing rates of BLA memory (*n* = 5 memory assemblies) and non-memory assemblies (*n* = 72 non-memory assemblies) during pre- or post-training sleep. Bottom panels, similar to top panels except showing BLA memory neurons (*n* = 16 memory neurons; *n* = 258 non-memory neurons). In box plots (**b** and **g**), the central lines indicate the median value, whereas the box edges and whiskers mark the interquartile ranges and limits, respectively. ****P* < 0.001, *****P* < 0.0001, two-tailed paired *t* test.

GLM decoder, and the remaining 50% were used for decoding. We used correlation coefficients between the real and predicted firing rates to indicate prediction power, resulting in a scale between −1 and 1 (Fig. 4d).

Our findings demonstrated that the dCA1 population spikes predicted BLA memory assembly firing rate during the post-, but not pre-training sleep (Fig. 4 d–f). This provides direct evidence of emerging weighted communications from dCA1 neurons to a selective BLA assembly in memory formation. In contrast, dCA1 population spikes had limited prediction on BLA non-memory neurons, although the prediction power on a small subset of them appeared to be high during both pre- and post-training sleep, which may reflect pre-existing dCA1–BLA communications in consolidating older memory traces (Fig. 4g).

## Emerging many-to-one weighted mapping underlies memory formation

Previous attempts to directly characterize dCA1 spike patterns that encode specific episodic events, such as experiencing shocks or air puffs[2,32], have not yielded a framework for understanding dCA1 information coding principles. Therefore, we employed a different approach to deduce the encoding principle of dCA1 neurons by analyzing their relationship with specific BLA assemblies. This approach leverages our ability to isolate a distinct BLA memory assembly that is involved in memory acquisition, consolidation, and retrieval (Fig. 2). Our results revealed that most dCA1 neurons exhibit BLA memory assembly-correlated activity during training and post-training sleep, but not pre-training sleep (Fig. 4a; Supplementary Figs. 2&3). This suggests the

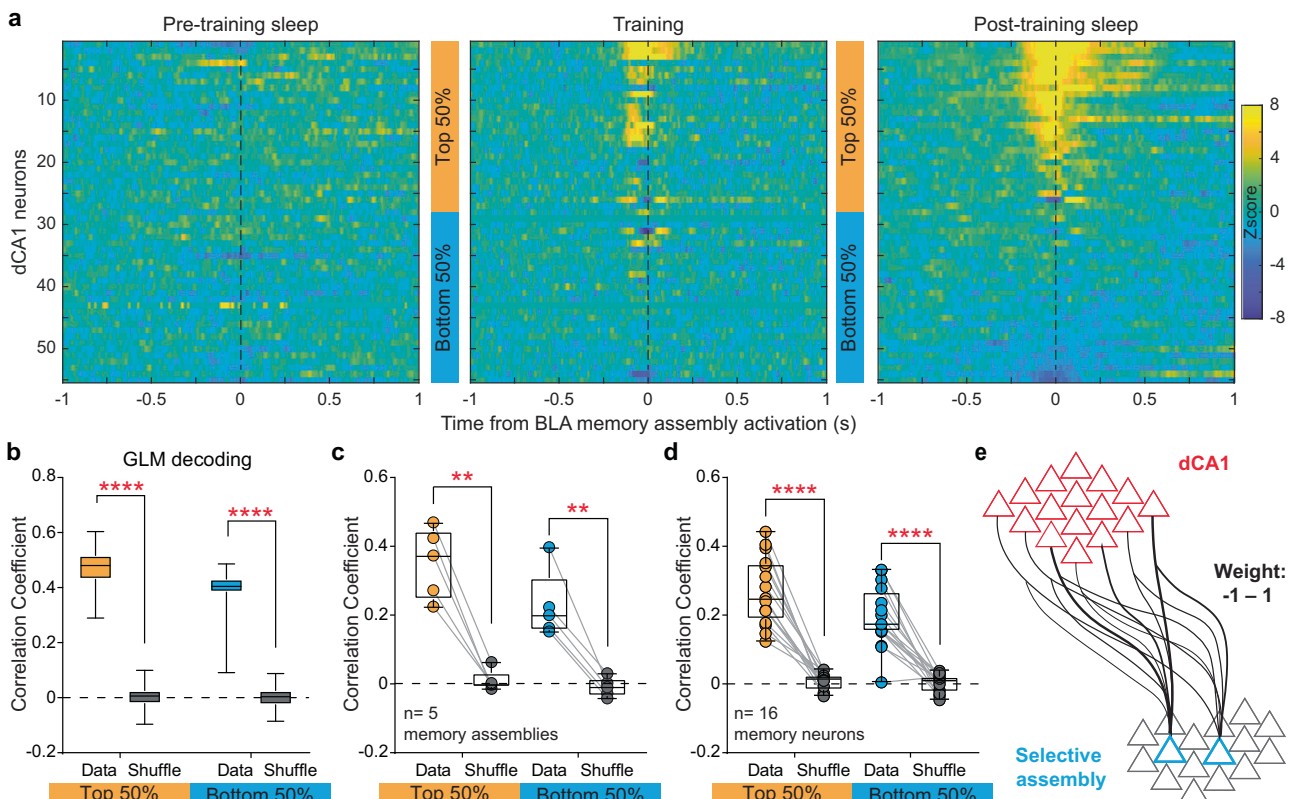

**Fig. 5 | Emerging many-to-one weighted mapping underlies memory formation. a** Cross-correlogram heatmaps between one representative BLA memory assembly and simultaneously recorded dCA1 neurons during the pre-training sleep (left), training (middle), and post-training sleep (right). Neurons are arranged in the same order across the three heatmaps. **b** Both the top 50% (corresponding to neurons #1–27 shown in A) and bottom 50% dCA1 neurons (#29–55) predicted the firing rates of the BLA assembly, in comparison to the shuffled data (repeated 100 times; $n = 100$). ****$P < 0.0001$; two-tailed paired $t$ test. **c, d** Both the top 50% and bottom 50% dCA1 neurons predicted the firing rates of individual BLA memory assemblies (**c**, $n = 5$ memory assemblies) and memory neurons (**d**, $n = 16$ memory neurons). In box plots, the central lines indicate the median value, whereas the box edges and whiskers mark the interquartile ranges and limits, respectively. **$P < 0.01$; ****$P < 0.0001$; two-tailed paired $t$ test. **e**, A proposed model of many-to-one weighted mapping from dCA1 to BLA.

involvement of the majority of dCA1 neurons compared to a small assembly of BLA neurons (5.3%) in memory acquisition and consolidation.

To further uncover the extent of dCA1 involvement in communication with the BLA, we divided the dCA1 neurons into subgroups based on their high or low activity in correlation with the BLA memory assembly (Fig. 5a). Our GLM decoding results showed that not only the higher-, but also lower-weight dCA1 neurons predicted the BLA assembly firing rate (Fig. 5 b&c; Supplementary Figs. 2&3). This suggests that decoding the BLA assembly activity involves contributions from many dCA1 neurons, rather than just a few. Taken together, we propose a model for memory formation: an emerging many-to-one weighted mapping from dCA1 neurons to a selective BLA assembly underlies the formation of a new memory (Fig. 5d).

In a follow-up analysis, we compared the spatial information coding properties between the higher- and lower-weight dCA1 neurons as classified above. Our results revealed that subsets of both groups exhibit place cell characteristics (Supplementary Fig. 6). We found no difference in spatial information coding capacity between the higher- and lower-weight dCA1 neurons (Supplementary Fig. 6). These results align with previous findings and support the notion that memory coding (non-spatial) and spatial coding are represented by partially overlapping groups of dCA1 neurons[33,34].

## Closed-loop optoinhibition during post-training sleep impairs memory

To determine if dCA1 ripple-coincided BLA activity is necessary for memory consolidation, we employed a closed-loop optoinhibition approach. We microinjected AAV-CaMKII-stGtACR2[35] into the BLA

bilaterally and then implanted two optic fibers slightly above the injection sites, which enabled later optoinhibition of BLA pyramidal neurons. Meanwhile, we implanted four tetrodes into the dCA1 to record ripple activity (Fig. 6a). After 2–3 weeks to allow viral expression, mice underwent a contextual fear conditioning procedure, followed by closed-loop, delayed, or no-stimulation of the BLA (Fig. 6b). This manipulation lasted two hours during post-training rest/sleep, similar to that reported previously[36,37].

Our analysis showed that large-amplitude dCA1 ripples had a greater correlation with BLA memory assembly activity (Fig. 2c; Supplementary Figs. 2&3). Therefore, we used a high threshold (8 s.d.) to trigger closed-loop optoinhibitions of the BLA, aiming to disrupt information flow into the BLA immediately after large ripple events (Fig. 6c). Our offline analysis confirmed that the closed-loop group mice received the optoinhibition within a short latency of ~10 ms after ripple detection, whereas the delayed group had an extended latency of ~160 ms (Fig. 6d). Behaviorally, the closed-loop group mice exhibited significantly reduced freezing compared to the delayed or no-stimulation groups, indicating impaired contextual fear memory (Fig. 6e). These findings suggest a crucial role of dCA1–BLA communication in memory formation.

## dCA1 communicates with BLA through a relay region

Anatomically, dCA1 and BLA are not directly connected by synapses[38,39]. Therefore, dCA1 potentially communicates with BLA through a relay region. This notion aligns with our finding that the dCA1-to-BLA information flow takes about 35 ms, indicating a disynaptic communication. It appears that only the entorhinal,

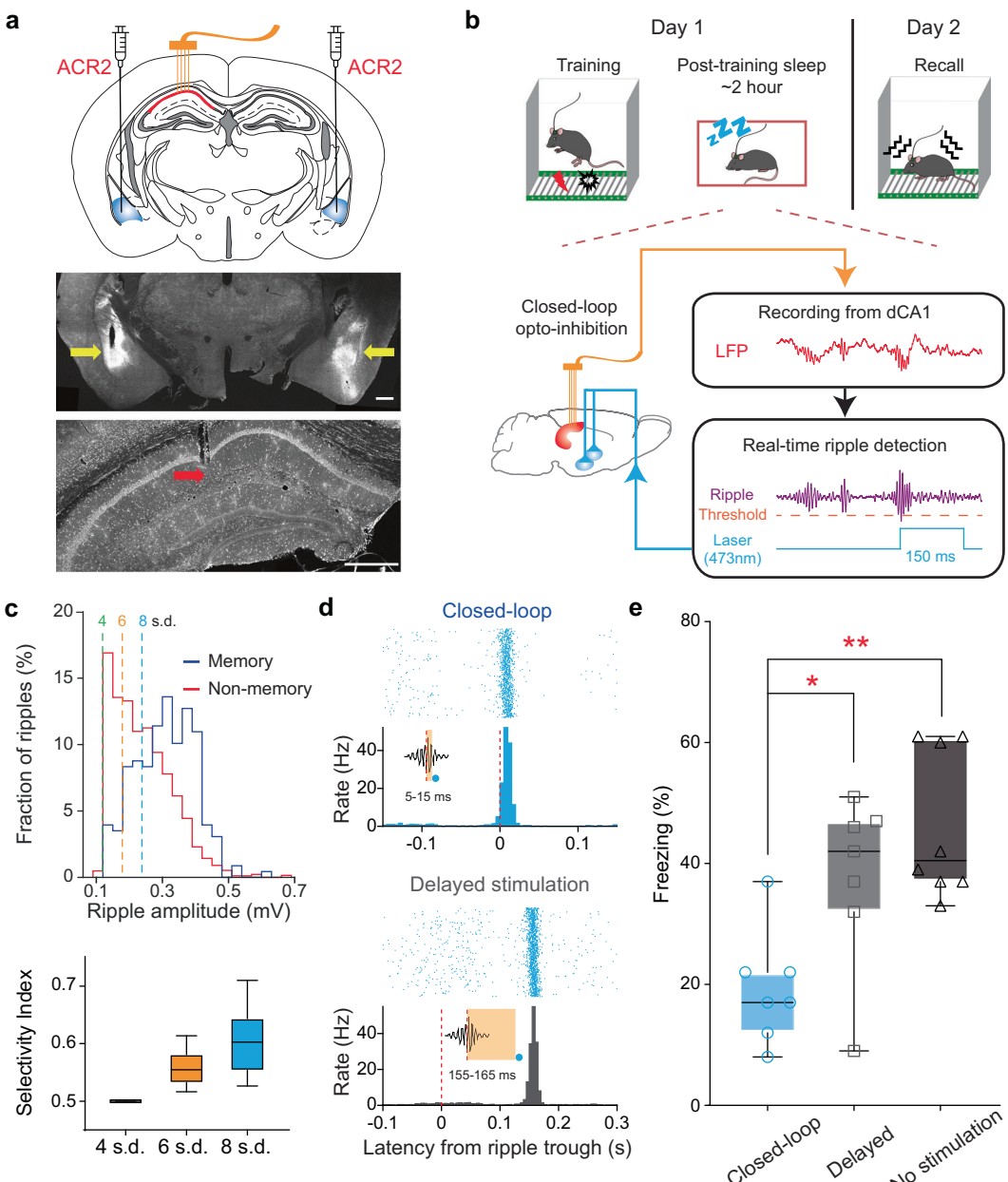

**Fig. 6 | Closed-loop optoinhibition during sleep impairs memory. a** Schematic of surgical procedure (top), and representative coronal sections showing viral expression mainly in the BLA (middle) and recording sites in the dCA1 (bottom). Scale bars, 0.5 mm. **b** Schematic of contextual fear memory procedure and closed-loop optoinhibition. **c** Ripples detected at higher threshold (8 vs. 6 vs. 4 s.d.) have higher selectivity of memory-associated ripples. Selectivity index = M/(M + N). M (or N) indicates the percentage of above-threshold ripples among memory-associated (or non-memory) ripples. Note that, despite the selectivity, only

11.5 ± 1.2% (mean ± s.e.m.) of the large-amplitude ripples ( >8 s.d.) were classified as memory-associated ripples ($n = 10$ mice). **d**, Optoinhibition latencies of the closed-loop group mice (top) and delayed-stimulation group (bottom). **e**, Closed-loop group mice ($n = 7$) shows impaired fear memory compared to delayed- ($n = 7$) or no-stimulation groups ($n = 8$, $P = 0.0013$, $F_{2, 19} = 9.583$, one-way ANOVA; *$P = 0.0303$, **$P = 0.0011$, Bonferroni post-hoc). In box plots (c&e), the central lines indicate the median value, whereas the box edges and whiskers mark the interquartile ranges and limits, respectively.

perirhinal, and ectorhinal cortices receive direct inputs from the dCA1 and project directly to the BLA, based on comprehensive anterograde and retrograde tracing studies shown in open-source databases, including the Mouse Connectome Project (Fig. 7a) and Allen Brain Atlas – Mouse Connectivity (Fig. 7b). Theoretically, information relayed from dCA1 to BLA could utilize a many→fewer→one or many→many→one principle (Fig. 7c). The former notion gains support in findings from immediate early gene studies, demonstrating that a large portion of dCA1 neurons (up to 50%) are activated during memory processing, whereas a moderate portion of entorhinal neurons (~25%), and only a small portion of BLA neurons (~10%) are activated in the

same process[16–18]. These differences are further amplified with decreasing neuron numbers, from ~400 k in dCA1, ~330 k in deep-layer entorhinal cortex, to only ~83 k in BLA of rats[40–43].

## Discussion

Our findings suggest that most dCA1 neurons concurrently index an episodic event by rapidly establishing weighted communications with a specific BLA assembly. We refer to this process as "many-to-one weighted mapping." The involvement of a significant portion of dCA1 neurons in encoding a specific event presents several theoretical advantages. Firstly, this mechanism offers immense encoding capacity due to the

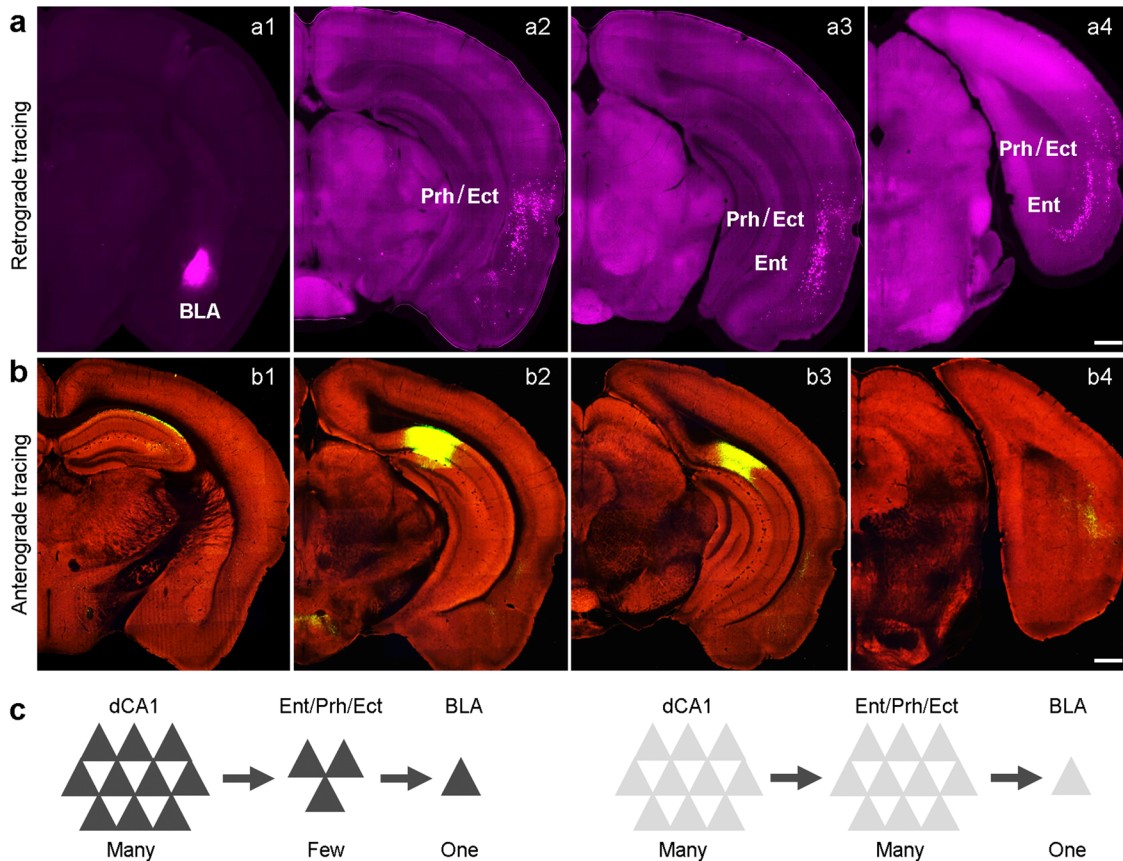

**Fig. 7 | dCA1 communicates with BLA through a relay region. a** Retrograde tracing of inputs to the BLA. Retrograde tracer CTB was microinjected into the BLA (a1). a2–a4, Retrograde labeling of neurons primarily in deep layers of entorhinal (Ent), perirhinal (Prh), and ectorhinal cortices (Ect). **b** Anterograde tracing of dCA1 efferents. Anterograde tracer (AAV-Syn-EGFP) was microinjected into the dCA1 and adjacent subiculum (b2). Anterograde projection is seen mainly in the Ent (b3) and Prh/Ect (b2–b4), but minimal in the BLA. **c** Two theoretical models of dCA1-to-BLA communication (many→fewer→one vs. many→many→one). Brain images are adapted from Mouse Connectome Project (**a**; https://cic.ini.usc.edu/) and Allen Brain Atlas (**b**; https://connectivity.brain-map.org/). Scale bars, 0.5 mm.

numerous potential combinations of dCA1 weighted mappings. Secondly, it provides robustness and redundancy, thereby enhancing single-trial learning and minimizing potential disruptions from background or external noises[9,44]. It is noteworthy that single-trial learning and memory formation are recognized as features of biological intelligence in comparison to long-training sessions of artificial intelligence[45]. On the other hand, the small proportion of participating BLA neurons (5.3%) may contribute to simplicity and efficiency in readout.

In essence, we propose that many-to-one weighted mappings are responsible for creating distinct memories within network representations. Specifically, when intermittent dCA1 firings occur that represent any significant mental or cognitive process spanning a few hundred milliseconds, they constitute an exclusive population activity pattern. This patterned firing enables the establishment of a unique weighted mapping from many dCA1 neurons to a specific BLA assembly that represents, in a primitive form, the same significant event, likely through rapid long-term potentiation[46]. In other words, distinct dCA1 population activity patterns communicate with corresponding BLA assemblies to rapidly index various episodic memories. It is possible that this dCA1 population activity pattern simultaneously communicates with multiple neuronal assemblies that are distributed across the brain through collateral projections[47–49]. This would allow dCA1 reactivations during ripples to bind distributed assemblies across brain regions to form a network representation of long-term memories. In support, we and others have shown that ripple-triggered optoinhibition of the BLA (Fig. 6) or the medial prefrontal cortex[50] impairs learning and memory formation.

Few studies have investigated the interactions between hippocampus and amygdala at the neuronal ensemble level during memory formation[2,3]. One study reported increased correlation between dCA1 neurons and selective BLA neurons during fear memory formation[2]. Our study extended this finding by showing that population spikes in dCA1 collectively predicted the firing rate of a selective BLA assembly during ripples. Another study found that the BLA assemblies are preconfigured before memory acquisition[3]. Our results confirmed this finding and further demonstrated that the activity of a unique BLA assembly was preceded by dCA1 ripples during post- but not pretraining sleep. We speculate that the recruitment of preexisting BLA assemblies to establish communication with the dCA1 enables rapid and efficient circuit mapping[29,51–53]. Together, an emerging hippocampus-to-amygdala communication appears to underline the acquisition and consolidation of new salient memories.

One outstanding question concerns how similar contextual fear experiences are encoded by the dCA1–BLA network. Prior research has shown that lesions of the dorsal hippocampus abolish the discrimination of similar contexts, indicating its crucial role in context discrimination[54]. It is plausible that different subpopulations of dCA1 neurons each encode a distinct context, albeit with a considerable amount of overlap between the dCA1 subpopulations (~30–42%), based on previous work[16,20]. This distinction appears to amplify upstream of CA1, where less- or even non-overlapping subpopulations of neurons within the CA3 and dentate gyrus are involved in distinguishing different contexts[16,20]. In contrast, we speculate that the same assembly of BLA neurons will be activated in multiple fear contexts. This notion is

supported by prior studies implicating the central role of the BLA in fear generalization[55,56]. Additionally, our preliminary findings revealed that many BLA memory neurons also responded to novel contexts after the contextual fear training (Supplementary Fig. 7), indicating that these BLA neurons exhibit generalized fear responses to multiple contexts.

In our experiments, we used relatively strong footshocks to induce salient memory, resulting in a many-to-one weighted mapping from dCA1 to BLA. Notably, the weights of correlation vary greatly across dCA1 neuron–BLA assembly pairs, forming a continuum spectrum from low to high values. We speculate that for less salient memories, the overall range of dCA1-BLA communication weights will be narrower (i.e., indicating a weaker association). Accordingly, the proportion of dCA1 neurons that significantly contribute to memory formation will also reduce.

One caveat of the current study is the lack of an experiment that directly tests our proposed model of many-to-one weighted mapping. Here, we propose two possible experiments for future investigation. The first experiment involves conducting retrograde tracing using transsynaptic viruses, such as the modified rabies virus[57] or pseudorabies virus[58]. Based on our many-to-one model, the infection of one or a few neurons within the BLA will result in exponential labeling of many neurons in upstream projection regions. The second experiment involves conducting high-resolution focal stimulation of multiple sites within a projection region while recording from the BLA[59]. If our model is correct, these focal stimulations will activate the same assembly of BLA neurons, albeit at varying levels of activation (i.e., weighted communication).

Our research also demonstrated that memory-associated dCA1 ripples are elongated, consistent with a recent study showing that prolonging dCA1 ripples after learning improves memory[28]. We found that the elongation of ripples is accompanied by an overall prolongation and enhancement of neuronal firings in the dCA1, with some neurons upregulating and others downregulating their activity. Additionally, our results revealed that memory-associated ripples were enlarged in amplitude, contrary to a previous study[28], which may reflect species differences between mice and rats[60]. Together, our findings suggest that elongated and enlarged dCA1 ripples signify the consolidation of newly acquired memories, while shorter and smaller ripples may indicate baseline activity or preconfigured dCA1 rigid motifs[29,52].

## Methods

### Mice

Male C57BL/6 mice were purchased from the Jackson Laboratory (stock #000664). Mice were 3–4 months old at the time of surgery; after surgery, they were singly housed in cages ($40 \times 20 \times 25$ cm) containing corn cob and cotton material and kept on a 12 h light/dark cycle with *ad libitum* access to food and water. Experimental procedures were approved by the Institutional Animal Care and Use Committees of Drexel University (protocol # LA-23-740) and were in accordance with the National Research Council *Guide for the Care and Use of Laboratory Animals*.

### Stereotaxic surgery

Surgery procedures were similar to that used in our lab[24]. In brief, mice were anesthetized with ketamine/xylazine mixture (~100/10 mg/kg, i.p.) and kept on a heating pad at 37 °C. For in vivo electrophysiology recording, mice received implantation of two electrode arrays (8–16 tetrodes each) into the BLA and dCA1, respectively[61]. For closed-loop optoinhibition, mice received microinjection of AAV viruses (AAV1-CKIIa-stGtACR2-FusionRed; 0.25 µl; ~$10^{13}$ GC/ml; *Addgene* 105669) and implantation of two optic fibers (diameter 200 µm) into the BLA bilaterally; meanwhile, they received implantation of 4 tetrodes into the dCA1 unilaterally. AAV viruses were microinjected through a syringe pump (*World Precision Instruments*) over 5 min (50 nL/min), with an addition of 5 min before removal of the injection needle (34 gauge,

beveled). The BLA coordinates were AP −1.7 mm, ML 3.4 mm, DV 3.9 mm; the dCA1 coordinates were AP −2.6 mm, ML 1.8 mm, and DV 1.1 mm.

### In vivo electrophysiology

Each tetrode consisted of four wires (90% platinum 10% iridium; 18 µm diameter; *California Fine Wire*). A microdrive was used to couple with the electrode bundle, similar to that used in our lab[24,26]. Neural signal was preamplified, digitized, and recorded using a *Blackrock Neurotech CerePlex* except one dataset that was recorded using a *Plexon* acquisition system; meanwhile, animals' behaviors were recorded. With the *Blackrock* system, the local field potentials (LFPs) were digitized at 2 kHz and filtered at 500 Hz low cut; spikes were digitized at 30 kHz and filtered between 600–6000 Hz. For the *Plexon* system, the LFPs were digitized at 1 kHz and filtered between 0.7–300 Hz; spikes were digitized at 40 kHz and filtered between 400–7000 Hz. The tetrode arrays were gradually lowered daily until we recorded clear ripples and a substantial number of neurons; otherwise, mice were excluded from the study. The recorded spikes were sorted using the MClust 3.5[24]; key datasets were manually verified using *Plexon* Offline Sorter. In total, spikes from 10 mice were used for analyses in this study; the neuron numbers in BLA and dCA1 were 58/55, 41/63, 36/52, 35/40, 35/18, 33/64, 31/41, 29/50, 29/38, and 14/15, respectively.

### Assembly detection by independent component analysis (ICA)

ICA was performed as described previously[27]. In brief, spike counts of each neuron were binned at 25 ms and z-scored to generate a neuronal population activity matrix (neurons × bins). Coactivity patterns were then extracted from this data matrix in two major steps. Firstly, the number of significant coactivation patterns was estimated by calculating the independent components (ICs) of the data matrix with variances above a threshold derived from an analytical probability function for uncorrelated data (Marchenko-Pastur distribution; Supplementary Fig. 1). Secondly, fastICA was used to extract the coactivity patterns from the projection of the data matrix into the subspace spanned by the significant ICs. In simple terms, this method first finds the significant ICs and then rotates them to match the ideal assembly patterns. These detected assembly patterns often were comprised of a small number of neurons with high weights, along with a larger group of neurons with low or zero weights. Neurons whose weights exceeded 2 s.d. from the mean weight of each assembly pattern were classified as members of an assembly, as described in previously[62,63].

The activation strength of each assembly was calculated by projecting the columns of the z-scored spike matrix onto the axis defined by the corresponding assembly pattern (Supplementary Fig. 1). The activation events were identified when the activation strength exceeded 5 s.d. from the mean activation strength of each assembly, as described previously[3]. Notably, using other thresholds, including 4 s.d. and 6 s.d., reached the same conclusions. To investigate the significance of activation event rates, we computed the activation strength of surrogate assemblies (500 surrogates for each assembly) generated by randomly permuting the data matrix. For all activation strength calculations, the assembly patterns extracted from post-training SWS sessions were utilized as templates.

### Ripples

Ripples were band-pass filtered between 100–250 Hz and ripple envelope was smoothed with a Gaussian kernel (s.d. = 4 ms)[64]. Ripple amplitudes were defined as the peak values of ripple envelopes. For analyses, we used amplitudes exceeding 5 s.d. above the mean, except for Fig. 3, where amplitudes exceeding 3 s.d. above the mean were analyzed. Ripple onsets and offsets were defined as the points where ripple amplitudes exceeded 1 s.d. above the mean before and after the corresponding ripple peaks (Fig. 3a). Ripple length was defined as the

duration between the onset and offset; only ripples longer than 20 ms were used for further analysis.

## Memory and nonmemory associated ripples

We defined BLA memory assemblies or neurons if they displayed robust activation during memory acquisition and ripple-modulated activation during post-, but not pre-training sleep (Fig. 2c). dCA1 ripples (recorded during the post-training sleep) were defined as memory-associated ripples if the corresponding BLA memory assembly (or neuron) exhibited high activation (2 s.d. above the median) within 150 ms of the ripple onset. The remaining ripples were defined as non-memory ripples (Fig. 3a).

## Ripple contents (related to Fig. 3c)

Firing difference score was defined as M | N = abs(M−N) / (M + N). M and N are the mean firing rates of each dCA1 neuron calculated ±100 ms within the onset of memory and non-memory associated ripples, respectively. M1 | M2 was defined as abs(M1−M2) / (M1 + M2), in which M1 and M2 are the mean firing rates of each dCA1 neuron calculated ±100 ms within the onset of two randomly divided groups of memory-associated ripples. N1 | N2 was similarly defined as M1 | M2 except for non-memory ripples.

## Ripple-modulated BLA assemblies and permutation test

Ripple modulation was determined by correlating the spikes of individual BLA assembly with respect to dCA1 ripple events using a peri-event analysis with a bin size of 5 ms. A Gaussian kernel (s.d. = 15 ms) was then applied to smooth the peri-ripple histogram. Assemblies that showed a difference from the baseline by a z-score of 3.3 or greater for three or more consecutive bins (within ± 150 ms from ripple onset) were defined as ripple-modulated assemblies, as described previously[26]. To determine if the proportion of dCA1 ripple-modulated BLA assemblies was greater than chance, we conducted a permutation test. Specifically, we randomly shuffled the timing of dCA1 ripple events 100 times followed by cross-correlation analyses to calculate the proportion of BLA assemblies modulated by the shuffled ripple events. Next, we performed a Chi-squared test to compare the proportions of BLA assemblies modulated by real and shuffled ripple events. Our results revealed a significant difference ($P < 0.05$) in each of the animals during post-training sleep and in most (7/10) of the animals during pre-training sleep.

## GLM decoding

We constructed generalized linear models (GLMs) with a log link function to predict spike counts of individual BLA assemblies or neurons during ripples based on population spike counts in dCA1 across specific time windows[31]. Ripples detected during pre- and post-training sleeps, and all dCA1 and BLA neurons were included for the analysis. Spike counts of each neuron or assembly were binned in 100-ms bins relative to ripple onset: −300 to −200 ms, −200 to −100 ms, −100 to 0 ms, 0 to 100 ms for dCA1, and −65 to 35 ms, 35 to 135 ms, 135 to 235 ms, 235 to 335 ms for BLA. We used dCA1 population spike counts in different time bins to predict the spike count of a single BLA assembly or neuron. We randomly partitioned the ripples into two equally sized datasets: one of them was used to train the GLM decoder, and the other was used for the test. For the test phase, the model derived from the training phase was applied to the dCA1 population spike data in the test set, yielding predictions for the predicted BLA spike counts across ripples. Lastly, we conducted correlation coefficient analysis between the predicted and real BLA spike counts to measure GLM decoding power on a scale from −1 to 1.

## Spatial information analysis

To quantify spatial information coding, firing rate maps of individual dCA1 neurons were generated in 1 × 1 cm spatial bins in NeuroExplorer and smoothed by a Gaussian filter (filter width, 5 bins) before being sent to MATLAB for further analyses. Spatial information content for each dCA1 unit was calculated using the formula:

$$Information\ content(bits/spike) = \sum_i p_i \frac{\lambda_i}{\lambda} \log_2 \frac{\lambda_i}{\lambda}, \qquad (1)$$

where $\lambda_i$ is the mean firing rate in the i-th bin, $\lambda$ is the overall mean firing rate, and $p_i$ is the probability of the animal's being in the i-th bin, as described previously[65]. Only units with peak firing rates higher than 0.4 Hz in any spatial bin were used for further analysis[66].

## Fear conditioning (in vivo recording)

The fear-conditioning chamber used in the experiment was a square chamber measuring 25 × 25 × 32 cm, with a 36-bar shock grid floor (*Med Associates*). The behaviors of the mice were recorded using either the *Blackrock Neurotech* NeuroMotive or *Plexon* CinePlex video system. During training, the mice were first allowed to explore the footshock chamber for ~3 minutes. They then received up to 10 mild footshocks (0.75 mA, 0.5 s), with a 2–3 min interval between shocks. Note that we used direct-current footshocks to minimize electromagnetic noise, so occasionally mice missed a shock if they stood on two positively or negatively charged grids. One minute after the last shock, the mice were returned to their home cages. Approximately 2 hours later, the mice were placed back in the footshock chamber for a 5-min contextual fear test. Subsequently, as a control test, mice were placed into a neutral chamber (40 cm in diameter, 35 cm in height) for 5 min. Neural activity was recorded continuously, including the pre-training sleep (1–2 hours), training (~0.5 hour), post-training sleep (1–2 hours), contextual-fear and neutral-chamber tests (5 min each).

## Fear conditioning (closed-loop optoinhibition)

Three groups of mice were used: 1) closed-loop; 2) delayed; and 3) no-stimulation groups. All mice were singly housed and received daily habituations of in vivo recording for 1–2 weeks. All mice underwent a contextual fear procedure that consisted of three footshocks (0.75 mA, 2 s; 1–1.5 min apart), similar to that described previously[67]. Freezing behaviors were automatically scored using *Med Associates* VideoFreeze[67]. To conduct closed-loop optoinhibition, we used the *Open Ephys* recording system and *Opto Engine* lasers. Only large dCA1 ripples (raw 100–250 Hz filtered traces) with peak amplitude exceeding 8 s.d.[68] were used to trigger bilateral optoinhibition of the BLA (0.5 mW; 150 ms), either immediately (closed-loop group) or after a delay of 150 ms (delayed-stimulation group).

## Histology

To mark the final recording sites, we made electrical lesions by passing 20-second, 10-μA currents through two or more tetrodes. Mice were deeply anesthetized and intracardially perfused with ice-cold PBS or saline, followed by 10% formalin. The brains were removed and post-fixed in formalin for at least 24 hours. The brains were sliced into coronal sections of 50-μm thickness using *Leica* vibratome. Sections from the dual-site recording mice were stained with cresyl violet for microscopic examination of electrode placements, whereas other sections were mounted with Mowiol mounting medium mixed with DAPI for microscopic fluorescent examination of viral vector expression and optical fiber placements.

## Statistics

Sample sizes were based on previous similar studies in our labs[24,26]. To determine firing-rate change during dCA1 ripples, the value that deviates from the mean by a z-score of >3.3 ($P < 0.001$) for at least three consecutive bins (bin = 5 ms) was considered significant. Other statistical analyses include Analysis of Variance (ANOVA) followed by post-hoc Bonferroni, Wilcoxon rank-sum test, and Student's *t* test. All

statistical tested are two-sided when applicable; *P* values of 0.05 or lower were considered significant.

## Reporting summary

Further information on research design is available in the Nature Portfolio Reporting Summary linked to this article.

## Data availability

Key dataset used in this study has been deposited to a public repository: https://figshare.com/articles/dataset/_b_Emerging_many-to-one_weighted_mapping_in_hippocampus-amygdala_network_underlies_memory_formation_b_/27104269 Source data are provided with this paper.

## Code availability

Key MATLAB scripts used in this study have been deposited to a public repository: https://github.com/DVWangLab.

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

## Acknowledgements
We thank Dr. Vitor Lopes-dos-Santos for discussions on the ICA analysis and Drs. Gideon Rothchild and Hualou Liang for discussions on the GLM decoding. We thank Dr. Wen-Jun Gao for comments on an earlier version. This work was supported by the National Institutes of Health grants R01MH119102 (D.V.W.) and F31MH134582 (A.F.H.).

## Author contributions
Conceptualization & Methodology: J.L., D.V.W.; Investigation: J.L., A.F.H., D.V.W.; Writing—original draft: J.L., D.V.W.; Writing—review & editing: J.L., A.F.H., D.V.W.

## Competing interests
The authors declare no competing interests.
