## [Peer Review File · Nature Communications]

REVIEWER COMMENTS

Reviewer #1 (Remarks to the Author):

In this study titled 'Emerging many-to-one weighted mapping in hippocampus-amygdala network underlies memory formation', authors study the relationship of basolateral amygdala (BLA) neural assemblies and dCA1 neurons during ripples in the acquisition and consolidation of fear memory. Authors detected the assemblies in BLA both during pre-training period, which persisted to the post training slow wave sleep and showed that some of them co-activated with the dCA1 ripples in the post training slow wave sleep. Furthermore, the ripple size and content (firing rates of some dCA1 neurons) were shown to be higher when they co-occurred with the BLA assemblies. The synchrony in the BLA assemblies and dCA1 neurons seems to demonstrate the active communication during the memory consolidation phase of the experiment. In another experiment, closed loop inactivation of ripples during post training sleep reduced the freezing behavior of the mice, indicating the impairment in memory consolidation.

In general, the results demonstrating how assemblies in BLA synchronize with the ripples and the dCA1 neurons after the experience of fear memory are impressive. Moreover, the authors' demonstration of the necessity of BLA activity during high strength ripples in the consolidation of fear memory implies the necessity of BLA dCA1 ripple co-ordination in the consolidation of fear memory. However, more evidence to support the claim that the synchronization/communication happening during the post SWS sleep was related to the consolidation of the fear memory and was not present before the experience in between pre-SWS BLA assemblies and ripples would be desirable.

Below are a few comments outlined in no particular order:

Line 60: Authors present an example of a BLA assembly which is modulated by ripples (1 out of 14). It is unclear how many out of 48 total post-SWS BLA assemblies were modulated by ripples. Although average activations are shown to be significantly greater than shuffle, it should be made clear whether the proportion (significantly modulated assembly out of all assemblies) was greater than expected by chance, both in pre and post SWS sleep.

Figure 1H&I: It is unclear why an example of an activation of an assembly was followed up by 10 memory neurons, and not all the significantly correlated assembly activations. Please clarify the difference between memory assembly and memory neurons. Also, indicate what were the proportion of memory neurons during pre and post SWS sleeps and whether the proportions are

significantly different from chance? It would be clearer to study the activations of all significant BLA assemblies following ripples in all animals rather than activations of the memory neurons.

Figure 4B&C: The transition from the prediction of activity of an example BLA assembly (figure 4B) to memory neurons (figure 4C) is unclear. It would be clearer to see the predictions of BLA assembly activities in all animals or for all assemblies rather than the memory neurons.

Line 62: It is unclear whether the “BLA memory assembly” was identified as memory encoding and/or consolidating assembly. If the assembly is to encode memory, the identification based on enhanced activation with post-SWS ripples seems to not be sufficient for such naming. But, it might rather be identified based on the actual encoding or consolidation of the freezing behavior (correlation of memory encoding BLA assemblies/ensemble during shock and post-SWS BLA assemblies similarities).

Line 78: The identification of Memory-associated ripples based on coinciding “BLA memory assembly” which were identified based on whether or not they increased their activities around ripples in the first place, seems to be problematic, because none of these were identified based on their similarity with the memory of events.

Figure 3 E&F: It is not clear how many BLA assemblies and/or neurons were decoded. Also, throughout the paper, the number of samples (data points in each group) used should be generally mentioned together with the statistical p-value for the proper interpretation of statistical reliability.

Figure 3G; It is unclear how pre-training BLA memory and non-memory neurons were determined. Please clarify if they were calculated based on assemblies detected in the pre-training BLA activities. Please clarify the proportions of such neurons and whether these were significant or not.

Figure 4: Authors need more evidence to convince that the memory encoding BLA assemblies emerge with the group of comodulated CA1 neurons. The evidence regarding the emergence is presented as diverse correlations of an example BLA assembly with groups of dCA1 neurons. The example BLA assembly seems to be detected during post training sleep. The predictive correlations of dCA1 neurons shown subsequently seem to belong to post SWS sleep. However, such correlations might already be there in between dCA1 neurons with the pre-sleep BLA assemblies during pre-training periods. Authors showed that the BLA assemblies in pre-SWS sleep have higher activation rate (see figure 1G) which was not statistically different from post SWS sleep BLA assemblies. The point to note here is that the authors did not present any evidence that such correlations emerge for all BLA assemblies (or the ones that are modulated by ripples as opposed to an example assembly) only after the shock exposure. Evidence in Figure S2 seems to show an

example of cell assembly following such a trend. So, whether or not the proportion of the detected cell assemblies following such a trend in both pre and post SWS sleep was significant needs to be thoroughly tested. Crucially, demonstrating the absence of such predictive correlations before the memory event (using pre SWS sleep assemblies and dCA1 neuronal firing spikes) might be helpful in making the case for emergence of specific and strong communication of memory between dCA1 neurons and BLA assemblies after fear memory exposure.

Figure 4B: The full extent of error bars and mean/median in shuffle aren't visible.

Figure 4C: Please clarify the reason for the exclusion of 2 mice from this analysis.

Line 125: "This suggests the involvement of a significant proportion of dCA1 neurons compared to a small assembly of BLA neurons in information coding and memory formation." Please clarify what 'comparison' in this particular text is referring to (BLA assembly neurons vs dCA1 assembly neurons or the proportion of CA1 neurons vs proportion of BLA neurons). Fig 4A being referred to only presents an example activity of dCA1 neurons around a BLA assembly and does not provide any comparison of proportions of assemblies or the neurons.

Line 185: "showed that the activity of these preexisting BLA assemblies was not driven by dCA1 ripples"; please clarify the statement, when Fig 1G shows that average activation of pre-SWS BLA assemblies were higher than the surrogates and not different from post-SWS periods.

Reviewer #2 (Remarks to the Author):

The manuscript by the authors entitled, "Emerging many-to-one weighted mapping in hippocampus-amygdala network underlies memory formation" examines the relationship between dCA1 and BLA neural activity in the acquisition of a contextual fear memory. The authors performed dual-site tetrode recordings of dCA1 and BLA neurons during pre-training sleep, contextual fear conditioning, post-training sleep, and during recall of the conditioned context. The major finding of this manuscript is the "many-to-one weighted mapping", where the firing of many dCA1 neurons during post-training sleep predicts the firing of a specific BLA ensemble. Using a closed-loop optoinhibition approach, the authors showed that BLA activity triggered by dCA1 ripples during post-training sleep was necessary for learning to occur. Overall, I think that the findings presented by the authors are interesting. That said, I would like the authors to address my comments below.

Major Comments

1. I would like to see an expanded Discussion section in which the authors consider how the many-to-one weighted mapping mechanism relates to more complicated situations and behavioral paradigms. While it is not my expectation that the authors necessarily address every question I pose in the two paragraphs below in their manuscript, I think a broader consideration in their manuscript of how the mechanism that they report relates to more complicated situations and behavioral paradigms would be of great general interest.

One question I have is related to how the many-to-one weighted mapping is affected by the strength of an acquired memory. If a milder contextual fear conditioning protocol was used that produced a weaker fear recall when experimental mice are re-exposed to the context, how would that affect the many-to-one weighted mapping mechanism that the authors describe? Would the population of dCA1 neurons that project to the BLA assembly be reduced or, alternatively, would there be a weaker association between dCA1 neural activity and BLA assembly neural activity?

Another question I have is how would two very similar but different episodic events that both produce fear of a context be encoded by the mechanism that the authors reported? I am thinking of a scenario in which a mouse experiences fear conditioning Context A and then fear conditioning in Context B. Would the same population of dCA1 neurons access two different BLA assemblies for these similar but different episodic events? Alternatively, would both episodic events be encoded by two different populations of dCA1 neurons that converge on the same BLA assembly?

2. I appreciate the authors' consideration regarding the anatomical connections between dCA1 and BLA (Lines 190-202, Supplemental Figure 3). However, I do not think that this consideration is sufficient because the authors do not address how the mechanisms that they propose in Figure S3C could be tested experimentally. One option that I will extend to the authors is that they add an additional experiment or experiments in their revised manuscript that tests the proposed mechanisms in Figure S3C (which I think would greatly increase the overall impact of the manuscript). As an alternative to conducting additional experiments, another option I will extend to the authors is that they expand their Discussion section to explicitly consider how the proposed mechanisms in Figure S3C could be tested experimentally.

Furthermore, do the authors expect any differences between how the many dCA1 neurons communicate with intermediary regions (i.e. entorhinal, perirhinal, and ectorhinal cortices) to reach the specific BLA assembly during memory consolidation? Finally, in addition to the figure legend, I think it should be very clear in Figure S3A-B that the images are adapted from the Mouse Connectome Project and Allen Brain Atlas. That said, I think it is enough to reference these images/data in the text instead of including them in the supplemental.

Minor Comments

1. Lines 30-32; Although the authors include references, it would be helpful if this part of the Introduction was expanded a bit to explain the “sequences” through which dCA1 encodes the where and when of memory. I think expanding the Introduction a bit here would be helpful to general audiences reading the manuscript.
2. Lines 72-73; The authors mention “This suggests that memory formation simply recruits preconfigured BLA assembles [3], although we observed a small swapping of membership in some assembles.” What are the authors referring to regarding the “small swapping of membership in some assembles”? Does this observation have any implications for the mechanism that the authors describe in the manuscript?
3. Lines 210-217; What was the titer of the virus that the authors microinjected into BLA? What was the speed of the microinjection? How long did the needle remain in the BLA after each injection?
4. Figure 1A; Scalebars should be defined in the figure or figure legend.
5. Figure 1F; Each graph should be clearly labeled #1 - #14 for each assembly, and/or the legend should indicate that the colored assembles in 1E correspond to the traces in 1F.
6. Figure 1I; Color of the bars on the whiskers on the boxplots does not appear consistent (some appear black and some appear gray).
7. Figure 2B; Color of axes on left graph do not appear uniformly black.
8. Figure 2C; Color of x-axis does not appear black.
9. Figure 2D; Color of tick marks on right graph do not appear black.
10. Figure 2F; It is unclear to me what is meant by “emerged activity” for the light blue ticks in the figure legend.
11. Figure 5A; Scalebars are needed for both histology images and should be defined.
12. Figure S3A-B; Scalebars should be in the images and should be defined.

Reviewer #3 (Remarks to the Author):

In the manuscript “Emerging many-to-one weighted mapping in hippocampus-amygdala network underlies memory formation”, the authors use a contextual form of conditioning, which is BLA and dCA1 dependent, to study circuit/ systems mechanisms underlying consolidation of a memory. I summarize the main findings of this manuscript as follows:

1. There is a small assembly of neurons in the BLA that their activity coincides with the dCA1 ripples. This correlation of activity emerges after the training. The authors call this assembly “BLA memory assembly”.

2. The ripples preceding the “BLA memory assembly” have a larger amplitude and last longer.

3. A large number of recorded dCA1 neurons show correlated activity with a small number of neurons in the “BLA memory assembly” (hence, the phrase “many-to-one”).

4. Inactivation of the BLA neurons following ripples of larger amplitude (see point 2), reduces the freezing response in the conditioning context (a proxy for testing memory recall).

Based on these findings, authors draw the conclusion that the large amplitude, long lasting dCA1 ripples that precede the activity of the “BLA memory assembly” are necessary for memory consolidation. However, as I detail in the section Major comments, the experiments in this manuscript cannot be used to make such a conclusion.

Major comments

1. A main claim of this manuscript is that “dCA1 ripple-coincided BLA activity is necessary for memory consolidation” (also, Figure 5). Closed-loop photoinhibition, as used here, does not settle the issue. It only shows a correlation. As the authors mentioned, there is no direct connection between the dCA1 and the BLA. The coinciding activity could as well be explained by a shared upstream input. To determine the cause-and-effect relationship between the two regions, they must eliminate large amplitude dCA1 ripples (ex. see Girardeau et al., 2009; Roux et al., 2017). Otherwise, the authors should be explicit that the photoinhibition only shows a correlation.

2. Following point 1, in lines 148-149, it is said "Our analysis showed that large-amplitude dCA1 ripples had a greater impact on memory consolidation (Figure 2B)". I don't see data supporting this, neither in Figure 2B nor in any other figure in the manuscript. What is shown is that the large-amplitude dCA1 ripples usually coincide with the activation “BLA memory assembly” and that this co-occurrence increases after training.

3. The authors propose that it is the larger dCA1 ripples which act for memory consolidation (“Our analysis showed that large-amplitude dCA1 ripples had a greater impact on memory

consolidation”). If this is the case, then closed-loop photoinhibition of the BLA following smaller ripples should have minimal effect on memory performance. This experiment is missing.

4. The authors assert that the effect of closed-loop inhibition on memory performance is related to "memory-associated" ripples. This is based on the observation that "memory-associated" ripples have larger amplitudes (Figure 2B and 5C). Then, they use a high threshold for ripple amplitude (8 s.d.) to trigger the closed-loop photoinhibition. If my reading of Figure 5C is correct, there still remains about 45% "non-memory"-associated ripples with amplitude larger than 8 s.d. Consequently, in their photoinhibition of the BLA following large-amplitude ripples, they target a considerable portion of "non-memory"-associated ripples. What makes the interpretation of this manipulation particularly difficult is that the raw numbers or the ratio of "memory-to-non-memory" ripple events with amplitudes larger than 8 s.d., are not reported. If, for example, the total number of "non-memory-associated" ripples is much higher, then even by setting a threshold of 8 s.d., the BLA is photoinhibited more often following "non-memory-associated" ripples.

5. It appears that most of the results are based on just one (or two?) "BLA memory assembly" made of 8 neurons. Is this from one or more mice? Can one propose a general rule ("many-to-one mapping") with such a few numbers?

6. Lines 108-109: "...the dCA1 population spikes predicted BLA memory assembly firing rate during the post-, but not pre-training sleep (Figure 3 D–F)." But, based on the way the criteria are set, isn't this expected (a bias in selection)? That is, already an assembly is called "memory assembly", because it shows dCA1 ripple-correlated activity (Figure 1E)? See lines, 62-63, and 66.

Minor comments

1. Line 72: More details about how the assemblies change before and after training should be given. The word "small" should be given a quantity.

2. Lines 81-82: The following sentence is not clear to me: "Moreover, these memory-associated ripples contained distinct contents, i.e., spikes of distinct dCA1 neurons (Figure 2C)." Could the authors explain what they mean by "distinct content" and "distinct CA1 neurons"?

3. Lines 108-109: It refers to Figure 3D-F for pre-training sleep data, but the data are not shown. Perhaps they refer to shuffle data, or they may refer to Figure 3G?

4. Line 235: It's not common to refer to Principal Components (PCs) explicitly in the context of Independent Component Analysis (ICA). It is proper to write "...calculating the independent components (PCs)"

5. Line 246: The choice of parameters (e.g., 2 standard deviations for classifying neurons as assembly members, 5 standard deviations for identifying activation events) need to be justified. Why were these specific thresholds chosen and how sensitive are the results to changes in these values?

6. Line 290: Why is it decided to use this criterion to identify place cells? I am not aware of a published work using such a metric; please provide a reference. Also, please provide details for statistical analyses (ex. see Roux et al., 2017).

7. Figure 1 (G & I): To avoid confusion, please adjust Panel G and I to have the same color for pre-SWS.

8. Line 427: Does "n=48" represent the number of assemblies? If so, please add the number of neurons?

9. Figure S1 (A): The 14 assemblies cover less than 50% of the variance in the data. Perhaps this is not the most effective way to detect assemblies. If the aim is to find assemblies of cells in the BLA that coincide with the dCA1 ripples, one can use selectivity analysis by comparing the activity of cells during ripples with activity at other time points.

10. Figure S2 (A). Please show the same plots for all the assemblies. This is to see if the increased responses during the training and the recall are specific to the "memory assembly".

11. The data shown for place cells are not directly relevant to the main theme of Figure 4. They could go to supplementary figures.

We thank the reviewers for their constructive comments. In our revision, we have added dual-site *in vivo* recording data from 5 additional animals, bringing the total number to 10, and conducted further analyses. Additionally, we have included comprehensive discussions to further elaborate the “many-to-one weighted mapping” mechanism. We believe these additions have greatly strengthened our conclusions.

Reviewer #1 (Remarks to the Author):

In this study titled ‘Emerging many-to-one weighted mapping in hippocampus-amygdala network underlies memory formation’, authors study the relationship of basolateral amygdala (BLA) neural assemblies and dCA1 neurons during ripples in the acquisition and consolidation of fear memory. Authors detected the assemblies in BLA both during pre-training period, which persisted to the post training slow wave sleep and showed that some of them co-activated with the dCA1 ripples in the post training slow wave sleep. Furthermore, the ripple size and content (firing rates of some dCA1 neurons) were shown to be higher when they co-occurred with the BLA assemblies. The synchrony in the BLA assemblies and dCA1 neurons seems to demonstrate the active communication during the memory consolidation phase of the experiment. In another experiment, closed loop inactivation of ripples during post training sleep reduced the freezing behavior of the mice, indicating the impairment in memory consolidation.

In general, the results demonstrating how assemblies in BLA synchronize with the ripples and the dCA1 neurons after the experience of fear memory are impressive. Moreover, the authors’ demonstration of the necessity of BLA activity during high strength ripples in the consolidation of fear memory implies the necessity of BLA dCA1 ripple co-ordination in the consolidation of fear memory. However, more evidence to support the claim that the synchronization/communication happening during the post SWS sleep was related to the consolidation of the fear memory and was not present before the experience in between pre-SWS BLA assemblies and ripples would be desirable.

We thank the reviewer for the positive comments and raising an important question regarding the specificity of hippocampal-amygdala communication in memory consolidation. This issue is discussed in detail below.

Below are a few comments outlined in no particular order:

Line 60: Authors present an example of a BLA assembly which is modulated by ripples (1 out of 14). It is unclear how many out of 48 total post-SWS BLA assemblies were modulated by ripples. Although average activations are shown to be significantly greater than shuffle, it should be made clear whether the proportion (significantly modulated assembly out of all assemblies) was greater than expected by chance, both in pre and post SWS sleep.

In our revised manuscript, we have included a supplementary figure (Suppl. Fig. 4) to show the activity of all BLA assemblies ($n = 88$) in relation to dCA1 ripples, both during pre- and post-training sleep. Notably, often one BLA assembly (or neuron) from each dataset showed robust dCA1 ripple-modulated activation during post-, but not pre-training sleep (Fig. 1f; Supplementary Figs. 2–4). Although there are other ripple-modulated BLA assemblies, they often exhibit decreased activity immediately after ripple events, or little modulation between pre- and post-training sleep (Fig. 1f; Supplementary Figs. 2–4).

To determine if the proportion of dCA1 ripple-modulated BLA assemblies was greater than chance, we conducted a permutation test. Specifically, we randomly shuffled the timing of dCA1 ripple events 100 times and conducted cross-correlation analyses to calculate the proportion of BLA assemblies modulated by the shuffled ripple events. An assembly was considered ripple modulated if the

correlogram peaks within 200 ms of ripple onset exceeded 3.3 s.d. above baseline ($P < 0.001$). Next, we performed a Chi-squared test to compare the proportions of BLA assemblies modulated by real and shuffled ripple events. Our results revealed a significant difference ($P < 0.05$) in each of the animals during post-training sleep and in most (7/10) of the animals during pre-training sleep. We now include these descriptions in the Methods section (lines #364–376).

Figure 1H&I: It is unclear why an example of an activation of an assembly was followed up by 10 memory neurons, and not all the significantly correlated assembly activations. Please clarify the difference between memory assembly and memory neurons. Also, indicate what were the proportion of memory neurons during pre and post SWS sleeps and whether the proportions are significantly different from chance? It would be clearer to study the activations of all significant BLA assemblies following ripples in all animals rather than activations of the memory neurons.

In our revised manuscript, we termed a BLA assembly as a “BLA memory assembly” if it displayed robust activation during memory acquisition and ripple-modulated activation during post-, but not pre-training sleep (Fig. 2). Individual neurons within each BLA memory assembly or single BLA neurons that met the above criteria were thus termed “BLA memory neurons.”

Given the challenge of dual-site *in vivo* recording and the small number of recorded “memory assemblies”, in our original submission, we only conducted statistical analysis on “memory neurons”. Since we added data from 5 additional animals during the revision, we now include statistical analyses for both “memory assemblies” and “memory neurons” (Suppl. Fig. 5)

To determine if the proportion of BLA memory neurons was greater than chance, we conducted a permutation test, similar to that described earlier. Specifically, we randomly shuffled the timing of dCA1 ripple events 100 times and conducted cross-correlation analyses to determine the proportion of BLA neurons modulated by the shuffled ripple events. Next, we performed a Chi-squared test to compare the proportions of BLA memory neurons identified by real and shuffled ripple events. Our results revealed a significant difference in each of the animals ($P < 0.05$).

Figure 4B&C: The transition from the prediction of activity of an example BLA assembly (figure 4B) to memory neurons (figure 4C) is unclear. It would be clearer to see the predictions of BLA assembly activities in all animals or for all assemblies rather than the memory neurons.

For the same reasons discussed above, we now include statistical analyses for both “memory assemblies” and “memory neurons” (Fig. 5).

Line 62: It is unclear whether the “BLA memory assembly” was identified as memory encoding and/or consolidating assembly. If the assembly is to encode memory, the identification based on enhanced activation with post-SWS ripples seems to not be sufficient for such naming. But, it might rather be identified based on the actual encoding or consolidation of the freezing behavior (correlation of memory encoding BLA assemblies/ensemble during shock and post-SWS BLA assemblies similarities).

Line 78: The identification of Memory-associated ripples based on coinciding “BLA memory assembly” which were identified based on whether or not they increased their activities around ripples in the first place, seems to be problematic, because none of these were identified based on their similarity with the memory of events.

In our revised manuscript, we termed a BLA assembly as a “BLA memory assembly” if it displayed robust activation during memory acquisition (encoding) and ripple-modulated activation during post-, but not pre-training sleep. We now include a new figure (Fig. 2).

Figure 3 E&F: It is not clear how many BLA assemblies and/or neurons were decoded. Also, throughout the paper, the number of samples (data points in each group) used should be generally mentioned together with the statistical p-value for the proper interpretation of statistical reliability.

We have updated the figures to include both assemblies and neurons (see Fig. 4). In our revised manuscripts, we have also included sample sizes and proper statistics in each figure or figure legend.

Figure 3G; It is unclear how pre-training BLA memory and non-memory neurons were determined. Please clarify if they were calculated based on assemblies detected in the pre-training BLA activities. Please clarify the proportions of such neurons and whether these were significant or not.

In our revised manuscript, we termed a BLA assembly as a “BLA memory assembly” if it displayed robust activation during memory acquisition and ripple-modulated activation during post-, but not pre-training sleep (Fig. 2). Individual neurons within each BLA memory assembly or single BLA neurons that met the above criteria were thus termed “BLA memory neurons.” In other words, pre-training and post-training “memory neurons” are the same neurons.

Overall, BLA memory neurons comprise ~5.3% (18/341) of all recorded BLA neurons. Based on the same permutation test discussed earlier, this proportion is significantly higher than chance ($P < 0.05$).

Figure 4: Authors need more evidence to convince that the memory encoding BLA assemblies emerge with the group of comodulated CA1 neurons. The evidence regarding the emergence is presented as diverse correlations of an example BLA assembly with groups of dCA1 neurons. The example BLA assembly seems to be detected during post training sleep. The predictive correlations of dCA1 neurons shown subsequently seem to belong to post SWS sleep. However, such correlations might already be there in between dCA1 neurons with the pre-sleep BLA assemblies during pre-training periods. Authors showed that the BLA assemblies in pre-SWS sleep have higher activation rate (see figure 1G) which was not statistically different from post SWS sleep BLA assemblies. The point to note here is that the authors did not present any evidence that such correlations emerge for all BLA assemblies (or the ones that are modulated by ripples as opposed to an example assembly) only after the shock exposure. Evidence in Figure S2 seems to show an example of cell assembly following such a trend. So, whether or not the proportion of the detected cell assemblies following such a trend in both pre and post SWS sleep was significant needs to be thoroughly tested. Crucially, demonstrating the absence of such predictive correlations before the memory event (using pre SWS sleep assemblies and dCA1 neuronal firing spikes) might be helpful in making the case for emergence of specific and strong communication of memory between dCA1 neurons and BLA assemblies after fear memory exposure.

We would like to first clarify that, prior to the contextual fear training, there is no (or minimal) correlated activity between dCA1 neurons and BLA memory assembly (Fig. 5a, left panel; Suppl. Figs. 2f&3f, left panels). However, there is a robust correlated activity between dCA1 neurons and BLA memory assembly during both memory encoding and consolidation phases (Fig. 5a, middle and right panels; Suppl. Figs. 2f&3f, middle and right panels). These results suggest the emergence of dCA1-to-BLA communication underlying memory formation.

Consistently, our GLM decoding results revealed that population dCA1 activity can predict the firing rates of BLA memory assembly (and memory neurons) during post-training SWS, but not during pre-training SWS (Fig. 4g). This further indicates the emergence of dCA1-to-BLA information flow underlying memory consolidation.

As for BLA non-memory neurons, although the activity of a subset of them can be decoded by population dCA1 activity, there is rarely an improvement during post-training SWS compared to pre-

training SWS. Overall, the prediction powers do not differ between pre- and post-training SWS for BLA non-memory neurons (Fig. 4g).

Figure 4B: The full extent of error bars and mean/median in shuffle aren't visible.

The means aren't visible because their values are very close to "0". We now clarify this in the figure legend.

Figure 4C: Please clarify the reason for the exclusion of 2 mice from this analysis.

We have clarified that the exclusion is due to the relatively small number of simultaneously recorded dCA1 neurons (<20), which renders the GLM decoding results unreliable. We now include this clarification in the revised manuscript (line #627).

Line 125: "This suggests the involvement of a significant proportion of dCA1 neurons compared to a small assembly of BLA neurons in information coding and memory formation." Please clarify what 'comparison' in this particular text is referring to (BLA assembly neurons vs dCA1 assembly neurons or the proportion of CA1 neurons vs proportion of BLA neurons). Fig 4A being referred to only presents an example activity of dCA1 neurons around a BLA assembly and does not provide any comparison of proportions of assemblies or the neurons.

We now specify that a small assembly of BLA neurons (5.3%; 18/341) and the majority of dCA1 neurons contribute to memory formation. And the reviewer is correct that Fig. 4A (now Fig. 5a) presents an example of emerging dCA1–BLA communication underlying memory formation. We now include two additional examples in our revised manuscript (Supplementary Figs. 2&3).

Line 185: "showed that the activity of these preexisting BLA assemblies was not driven by dCA1 ripples"; please clarify the statement, when Fig 1G shows that average activation of pre-SWS BLA assemblies were higher than the surrogates and not different from post-SWS periods.

We concluded that the preexisting BLA memory assemblies were not always driven by dCA1 ripples, because their activity was not correlated with dCA1 ripples (or dCA1 neurons) during pre-training SWS (Fig. 1f; Fig. 5a; Suppl. Figs. 2&3).

Regarding the observation that the average activation of pre-SWS BLA assemblies was higher than that of the surrogates, this mainly indicates preexisting assembly activity prior to contextual fear training. In other words, memory formation primarily recruits preconfigured or preexisting BLA assemblies, rather than forming new assemblies, a notion consistent with recent findings (Miyawaki & Mizuseki, *Nat. Commun.* 2022). We now include these discussions in the revised manuscript (lines #103–111).

Reviewer #2 (Remarks to the Author):

The manuscript by the authors entitled, “Emerging many-to-one weighted mapping in hippocampus-amygdala network underlies memory formation” examines the relationship between dCA1 and BLA neural activity in the acquisition of a contextual fear memory. The authors performed dual-site tetrode recordings of dCA1 and BLA neurons during pre-training sleep, contextual fear conditioning, post-training sleep, and during recall of the conditioned context. The major finding of this manuscript is the “many-to-one weighted mapping”, where the firing of many dCA1 neurons during post-training sleep predicts the firing of a specific BLA ensemble. Using a closed-loop optoinhibition approach, the authors showed that BLA activity triggered by dCA1 ripples during post-training sleep was necessary for learning to occur. Overall, I think that the findings presented by the authors are interesting. That said, I would like the authors to address my comments below.

Major Comments

1. I would like to see an expanded Discussion section in which the authors consider how the many-to-one weighted mapping mechanism relates to more complicated situations and behavioral paradigms. While it is not my expectation that the authors necessarily address every question I pose in the two paragraphs below in their manuscript, I think a broader consideration in their manuscript of how the mechanism that they report relates to more complicated situations and behavioral paradigms would be of great general interest.

We appreciate the reviewer’s suggestion to expand our Discussion section by relating our results to more complex learning situations, which would strengthen the impact our manuscript. We now include these discussions in our revised manuscript, as detailed below.

One question I have is related to how the many-to-one weighted mapping is affected by the strength of an acquired memory. If a milder contextual fear conditioning protocol was used that produced a weaker fear recall when experimental mice are re-exposed to the context, how would that affect the many-to-one weighted mapping mechanism that the authors describe? Would the population of dCA1 neurons that project to the BLA assembly be reduced or, alternatively, would there be a weaker association between dCA1 neural activity and BLA assembly neural activity?

We appreciate these thoughtful questions. While conducting new experiments with milder shocks would help address them, we have not pursued this for two main reasons. Firstly, our revision process has already been lengthy – nearly a year – due to the complex nature of dual-site *in vivo* recording (64–128 channels) and the inclusion of new data from an additional 5 animals. Further experiments would likely cause additional delays in publication. Secondly, we are concerned that milder shocks may result in a lower success rate in contextual fear learning, further reducing our chance of acquiring key neural dataset. Nevertheless, based on our current findings, we’d like to provide perspectives on these questions.

Our “many-to-one mapping” results show that many dCA1 neurons communicate with one BLA assembly, although the weights of communication vary greatly across dCA1 neuron–BLA assembly pairs. In fact, these weights form a continuum spectrum from low to high values (Fig. 5A; Suppl. Fig. 2f; Suppl. Fig. 3f), suggesting that distinct dCA1 neurons contribute different weights to the consolidation of a new memory. We speculate that, for “weaker memories”, the overall range of dCA1–BLA communication weights will be narrower (i.e., indicating a weaker association). As a result, the proportion of dCA1 neurons that significantly contribute to memory consolidation will also reduce. We have added these discussions in our revised manuscript (lines #264–269).

Another question I have is how would two very similar but different episodic events that both produce fear of a context be encoded by the mechanism that the authors reported? I am thinking of a scenario in which a mouse experiences fear conditioning Context A and then fear conditioning in Context B.

Would the same population of dCA1 neurons access two different BLA assemblies for these similar but different episodic events? Alternatively, would both episodic events be encoded by two different populations of dCA1 neurons that converge on the same BLA assembly?

This is a terrific question; unfortunately, we do not have data to directly address it. Below are our speculations.

Prior research has shown that lesions of the dorsal hippocampus abolish the discrimination of similar contexts, indicating its crucial role in context discrimination (Frankland, P.W., et al, *Behav Neurosci*, 1998). Therefore, it is plausible that different subpopulations of dCA1 neurons each encode a distinct context, albeit with a considerable amount of overlap between the dCA1 subpopulations (~30–42%), based on prior work (Leutgeb et al, *Science* 2004; Ramirez et al, *Science* 2013).

In contrast, we speculate that the same assembly of BLA neurons will be activated in multiple fear contexts. This notion is supported by prior studies implicating the central role of the BLA in fear generalization. In support, our preliminary findings revealed that many BLA memory neurons also responded to novel contexts after the contextual fear training (Supplementary Fig. 7), indicating that these BLA neurons exhibit generalized responses to multiple contexts.

Together, these findings suggest that the BLA assembly likely does not distinguish between contexts; rather, distinct subpopulations of dCA1 neurons encoding different contexts converge on the BLA assembly. We have added these discussions in our revised manuscript (lines #251–263).

2. I appreciate the authors' consideration regarding the anatomical connections between dCA1 and BLA (Lines 190-202, Supplemental Figure 3). However, I do not think that this consideration is sufficient because the authors do not address how the mechanisms that they propose in Figure S3C could be tested experimentally. One option that I will extend to the authors is that they add an additional experiment or experiments in their revised manuscript that tests the proposed mechanisms in Figure S3C (which I think would greatly increase the overall impact of the manuscript). As an alternative to conducting additional experiments, another option I will extend to the authors is that they expand their Discussion section to explicitly consider how the proposed mechanisms in Figure S3C could be tested experimentally.

We now discuss how our proposed “many-to-one” model could be tested experimentally (lines #270–279; see below).

“One caveat of the current study is the lack of an experiment that directly tests our proposed model of many-to-one weighted mapping. Here, we propose two possible experiments for future investigation. The first experiment involves conducting retrograde tracing using transsynaptic viruses, such as the modified rabies virus or pseudorabies virus. Based on our many-to-one model, the infection of one or a few neurons within the BLA will result in exponential labeling of many neurons in upstream projection regions. The second experiment involves conducting high-resolution focal stimulation of multiple sites within a projection region while recording from the BLA. If our model is correct, these focal stimulations will activate the same assembly of BLA neurons, albeit at varying levels of activation (i.e., weighted communication).” This same design could be further implemented to test relay region contributions once said region has been identified.

Furthermore, do the authors expect any differences between how the many dCA1 neurons communicate with intermediary regions (i.e. entorhinal, perirhinal, and ectorhinal cortices) to reach the specific BLA assembly during memory consolidation? Finally, in addition to the figure legend, I think it should be very clear in Figure S3A-B that the images are adapted from the Mouse Connectome Project and Allen Brain Atlas. That said, I think it is enough to reference these images/data in the text instead of including them in the supplemental.

We thank the reviewer for this suggestion. We now move this supplementary figure to the main text as Fig. 7. We also specified that these images are adapted from the Mouse Connectome Project and Allen Brain Atlas.

Regarding the potential differences in communication between dCA1 neurons and intermediary region assemblies, this is a terrific question. We speculate that a few assemblies in the intermediary region will be involved in encoding details or aspects of the contextual fear memory, such as distinguishing different contexts. This contrasts with BLA memory assembly, which encodes general fear regardless of the context.

Minor Comments

1. Lines 30-32; Although the authors include references, it would be helpful if this part of the Introduction was expanded a bit to explain the “sequences” through which dCA1 encodes the where and when of memory. I think expanding the Introduction a bit here would be helpful to general audiences reading the manuscript.

We now add explanations regarding how dCA1 encodes place and time by sequences (lines #33–34).

2. Lines 72-73; The authors mention “This suggests that memory formation simply recruits preconfigured BLA assemblies [3], although we observed a small swapping of membership in some assemblies.” What are the authors referring to regarding the “small swapping of membership in some assemblies”? Does this observation have any implications for the mechanism that the authors describe in the manuscript?

We have conducted additional analyses and clarified our main conclusions (Suppl. Fig. 1). Overall, our results support the notion that memory formation primarily recruits preconfigured or preexisting BLA assemblies, rather than forming new assemblies, a notion consistent with recent findings (Miyawaki & Mizuseki, *Nat. Commun.* 2022).

3. Lines 210-217; What was the titer of the virus that the authors microinjected into BLA? What was the speed of the microinjection? How long did the needle remain in the BLA after each injection?

We now include AAV and surgery details in our revised manuscript (see Methods: Stereotaxic surgery).

4. Figure 1A; Scalebars should be defined in the figure or figure legend.
Done.

5. Figure 1F; Each graph should be clearly labeled #1 - #14 for each assembly, and/or the legend should indicate that the colored assemblies in 1E correspond to the traces in 1F.
Done.

6. Figure 1I; Color of the bars on the whiskers on the boxplots does not appear consistent (some appear black and some appear gray).

It is all black now.

7. Figure 2B; Color of axes on left graph do not appear uniformly black.
It is all black now.

8. Figure 2C; Color of x-axis does not appear black.

It is black now.

9. Figure 2D; Color of tick marks on right graph do not appear black.

It is black now.

10. Figure 2F; It is unclear to me what is meant by “emerged activity” for the light blue ticks in the figure legend.

We have slightly revised this figure to avoid confusion.

11. Figure 5A; Scalebars are needed for both histology images and should be defined.

Done.

12. Figure S3A-B; Scalebars should be in the images and should be defined.

Done.

Reviewer #3 (Remarks to the Author):

In the manuscript "*Emerging many-to-one weighted mapping in hippocampus-amygdala network underlies memory formation*", the authors use a contextual form of conditioning, which is BLA and dCA1 dependent, to study circuit/ systems mechanisms underlying consolidation of a memory. I summarize the main findings of this manuscript as follows:

1. There is a small assembly of neurons in the BLA that their activity coincides with the dCA1 ripples. This correlation of activity emerges after the training. The authors call this assembly "BLA memory assembly".
2. The ripples preceding the "BLA memory assembly" have a larger amplitude and last longer.
3. A large number of recorded dCA1 neurons show correlated activity with a small number of neurons in the "BLA memory assembly" (hence, the phrase "many-to-one").
4. Inactivation of the BLA neurons following ripples of larger amplitude (see point 2), reduces the freezing response in the conditioning context (a proxy for testing memory recall).

Based on these findings, authors draw the conclusion that the large amplitude, long lasting dCA1 ripples that precede the activity of the "BLA memory assembly" are necessary for memory consolidation. However, as I detail in the section **Major comments**, the experiments in this manuscript cannot be used to make such a conclusion.

Major comments

1. A main claim of this manuscript is that "*dCA1 ripple-coincided BLA activity is necessary for memory consolidation*" (also, Figure 5). Closed-loop photoinhibition, as used here, does not settle the issue. It only shows a correlation. As the authors mentioned, there is no direct connection between the dCA1 and the BLA. The coinciding activity could as well be explained by a shared upstream input. To determine the cause-and-effect relationship between the two regions, they must eliminate large amplitude dCA1 ripples (ex. see Girardeau et al., 2009; Roux et al., 2017). Otherwise, the authors should be explicit that the photoinhibition only shows a correlation.

We thank the reviewer for bringing the causality vs. correlation issue to our attention. We now have toned down our claim to simply conclude on the important role (rather than causal role) of BLA activity in memory consolidation. Based on current results, it is unlikely that dCA1 ripples and the BLA assembly were driven by shared upstream inputs, because they did not co-activate in synchrony. Instead, dCA1 ripple activity preceded BLA assembly activation by approximately 35 ms (Fig. 4a). Additionally, our GLM decoding results indicated that dCA1 population activity best predicted the firing rate of the BLA assembly after a delay of 35 ms (Fig. 4f). Therefore, a more likely explanation is that dCA1 ripple influences BLA activity rather than coactivating with the BLA assembly.

2. Following point 1, in lines 148-149, it is said "*Our analysis showed that large-amplitude dCA1 ripples had a greater impact on memory consolidation (Figure 2B)*". I don't see data supporting this, neither in Figure 2B nor in any other figure in the manuscript. What is shown is that the large-amplitude dCA1 ripples usually coincide with the activation "BLA memory assembly" and that this co-occurrence increases after training.

The reviewer correctly pointed out that the activity of large-amplitude dCA1 ripples correlates with BLA memory assembly activity. We now highlight the correlation rather than causation.

3. The authors propose that it is the larger dCA1 ripples which act for memory consolidation ("*Our analysis showed that large-amplitude dCA1 ripples had a greater impact on memory consolidation*"). If this is the case, then closed-loop photoinhibition of the BLA following smaller ripples should have minimal effect on memory performance. This experiment is missing.

The suggested experiment is interesting but technically challenging, potentially not possible with currently available closed-loop software.

Disrupting small ripples while sparing large ones would require considerable computing time, in the range of tens of milliseconds at a minimum. This estimation is based on the duration of half-ripple length (~15–50 ms), as the calculation of ripple amplitude requires the processing of at least half of individual ripple events. Unfortunately, this delay (~50 ms) goes against our goal of real-time closed-loop manipulation, which requires minimal delay in the range of milliseconds.

Accordingly, we have now revised our claim to highlight the correlation rather than causation.

4. The authors assert that the effect of closed-loop inhibition on memory performance is related to "memory-associated" ripples. This is based on the observation that "memory-associated" ripples have larger amplitudes (Figure 2B and 5C). Then, they use a high threshold for ripple amplitude (8 s.d.) to trigger the closed-loop photoinhibition. If my reading of Figure 5C is correct, there still remains about 45% "non-memory"-associated ripples with amplitude larger than 8 s.d. Consequently, in their photoinhibition of the BLA following large-amplitude ripples, they target a considerable portion of "non-memory"-associated ripples. What makes the interpretation of this manipulation particularly difficult is that the raw numbers or the ratio of "memory-to-non-memory" ripple events with amplitudes larger than 8 s.d., are not reported. If, for example, the total number of "non-memory-associated" ripples is much higher, then even by setting a threshold of 8 s.d., the BLA is photoinhibited more often following "non-memory-associated" ripples.

We thank the reviewer for bringing this issue to our attention. We agree that our closed-loop manipulation affects both memory and non-memory ripples. And the reviewer is correct that there is a significant amount of non-memory ripples, even with a threshold of 8 s.d. On average, $10.1 \pm 0.6\%$ (mean \pm s.e.m.) of the large-amplitude ripples (i.e., amplitude >8 s.d.) were characterized as memory-associated ripples. We now report this qualification in the revised manuscript (lines #655–656).

Accordingly, we have revised our original claim to simply highlight the crucial role of large-amplitude ripples (rather than memory-associated ripples) in memory consolidation.

5. It appears that most of the results are based on just one (or two?) "BLA memory assembly" made of 8 neurons. Is this from one or more mice? Can one propose a general rule ("many-to-one mapping") with such a few numbers?

In our original submission, we included dual-site *in vivo* data from 5 mice. In our revision, we have added data from 5 additional animals, bringing the total number to 10, and conducted further analyses. We believe these additions have greatly strengthened our conclusions.

For transparency, we have added two supplementary figures to present examples from two additional animals (Supplementary Figs. 2&3).

6. Lines 108-109: "...the dCA1 population spikes predicted BLA memory assembly firing rate during the post-, but not pre-training sleep (Figure 3 D–F)." But, based on the way the criteria are set, isn't this expected (a bias in selection)? That is, already an assembly is called "memory assembly", because it shows dCA1 ripple-correlated activity (Figure 1E)? See lines, 62-63, and 66.

We thank the reviewer for bringing this issue to our attention. Here we would like to clarify that the two approaches complement each other in determining the relationship between dCA1 neurons and BLA memory assembly in memory consolidation.

Specifically, our criteria to identify a BLA memory assembly are based on its *averaged* correlations with dCA1 ripples (and responses during memory acquisition training). In contrast, the GLM decoding is based on *single-trial* relationships between population dCA1 neurons and a BLA assembly across individual ripple events. Although good correlation often leads to good decoding, it is not guaranteed. In fact, if the two regions shared an upstream input instead of a dCA1-to-BLA information flow, our GLM analysis decoding would perform poorly. Additionally, the ability for the GLM to only predict post-sleep activity demonstrates a unique feature of this communication that is only present following learning. If the GLM results were simply due to the coincidental activation of ripples and BLA memory assemblies, one might expect to see similar results across pre- and post-training sleep.

Minor comments

1. Line 72: More details about how the assemblies change before and after training should be given. The word "small" should be given a quantity.

We have conducted further analysis and included a revised Supplementary Fig. 1.

2. Lines 81-82: The following sentence is not clear to me: "*Moreover, these memory-associated ripples contained distinct contents, i.e., spikes of distinct dCA1 neurons (Figure 2C).*" Could the authors explain what they mean by "*distinct content*" and "*distinct CA1 neurons*"?

We have revised the sentence and added details in Methods: Ripple contents (lines: #358–363).

3. Lines 108-109: It refers to Figure 3D-F for pre-training sleep data, but the data are not shown. Perhaps they refer to shuffle data, or they may refer to Figure 3G?

Yes, the pre-training data is shown in Fig. 3G (now Fig. 4g). We have corrected this.

4. Line 235: It's not common to refer to Principal Components (PCs) explicitly in the context of Independent Component Analysis (ICA). It is proper to write "*...calculating the independent components (PCs)*"

We thank the reviewer for the comments and have revised the sentence as suggested.

5. Line 246: The choice of parameters (e.g., 2 standard deviations for classifying neurons as assembly members, 5 standard deviations for identifying activation events) need to be justified. Why were these specific thresholds chosen and how sensitive are the results to changes in these values?

The choice of 2SD is consistent with recent studies employing similar approaches (El-Gaby et al, *Nature Neuroscience* 2021; van de Ven et al, *Neuron* 2016). This threshold is suitable for classifying BLA assembly membership, as additional criteria based on BLA neuronal responses during memory acquisition/retrieval largely identified the same neurons as assemblies (Fig. 2).

Similarly, the choice of 5SD for identifying activation events is based on established research (Miyawaki & Mizuseki, *Nat. Commun.* 2022). Using alternative thresholds, such as 4SD or 6SD, does not alter our main conclusions. We have added this justification to our Methods section.

6. Line 290: Why is it decided to use this criterion to identify place cells? I am not aware of a published work using such a metric; please provide a reference. Also, please provide details for statistical analyses (ex. see Roux et al., 2017).

We opt to include place cell analysis based on information content (bits/spike) and cited relevant publication (Skaggs et al, 1996; Roux et al., 2017). Please see Supplementary Fig. 6 and Methods: Spatial information analysis (lines: #390–397).

7. Figure 1 (G & I): To avoid confusion, please adjust Panel G and I to have the same color for pre-SWS.

Done.

8. Line 427: Does "n=48" represent the number of assemblies? If so, please add the number of neurons?

Yes. We now specify that each assembly comprises 2–6 neurons (line #75–77). For representatives, please see Fig. 1d; Suppl. Fig. 2a, and Suppl. Fig. 3a.

9. Figure S1 (A): The 14 assemblies cover less than 50% of the variance in the data. Perhaps this is not the most effective way to detect assemblies. If the aim is to find assemblies of cells in the BLA that coincide with the dCA1 ripples, one can use selectivity analysis by comparing the activity of cells during ripples with activity at other time points.

Our estimation of the number of assemblies is based on the Marchenko-Pastur distribution, which is detailed in this well-cited paper (Lopes-dos-Santos et al, 2013). Since its publication, this method has been employed by many reputable labs in the field, including the Buzsaki Lab (Fernandez-Ruiz et al, *Science* 2021; Huszar et al, *Nature Neuroscience* 2022), the Dupret Lab (van de Ven et al, *Neuron* 2016; El-Gaby et al, *Nature Neuroscience* 2021), and the Losonczy Lab (Grosmark et al, *Nature Neuroscience* 2021), among others. While there may be a more effective way to detect assemblies, our current method reflects the common standard in the field at this time.

10. Figure S2 (A). Please show the same plots for all the assemblies. This is to see if the increased responses during the training and the recall are specific to the “memory assembly”.

Please see our new Fig. 2.

11. The data shown for place cells are not directly relevant to the main theme of Figure 4. They could go to supplementary figures.

We have moved this set of results to Supplementary Fig. 6.

REVIEWERS' COMMENTS

Reviewer #1 (Remarks to the Author):

The authors in their revision were able to address most of the concerns that I have raised. I think the revision has strengthened and improved the paper in several ways.

- 1) The inclusion of additional data in their analyses helped make the results more reliable.
- 2) The identification of the preconfiguration in the BLA assemblies that were later recruited with the hippocampal ripples in the post-sleep helped clarify the inter-regional consolidation mechanism.
- 3) The toning down of the claims regarding causation have strengthened the evidence based interpretation of the results. Speculation about the causation and proposal of new experiments to test the causation seem to be valid discussions based on their findings.

Except for the following three minor comments, I have no further comments:

- 1) Line 368; The authors state “score of 3.3 or greater for three or more consecutive bins (within \pm 150 ms from ripple onset) were defined as ripple-modulated assemblies”, but rebuttal text says, “An assembly was considered ripple modulated if the correlogram peaks within 200 ms of ripple onset exceeded 3.3 s.d. above baseline ($P < 0.001$)”, please clarify the inconsistency. Also, please mention the criteria for selecting the BLA ripple modulated neurons (whether it is the same or different to the BLA ripple modulated assemblies).
- 2) In the rebuttal authors state, “For the same reasons discussed above, we now include statistical analyses for both “memory assemblies” and “memory neurons” (Fig. 5).” Fig 5 does not seem to have that information. I can find it in Fig 4 though. Please clarify.
- 3) Some of the key references are missing and should be added when discussing cell assembly detection and experience-dependent reactivation (Peyrache et al., 2009; Farooq et al. 2019) and network preconfiguration and neuronal selection (Dragoi and Tonegawa, 2011; Dragoi, 2024).

Reviewer #2 (Remarks to the Author):

In the revised manuscript, the authors thoroughly addressed my previous comments. I was pleased by the expanded Discussion section, increased N for experimental animals, revised main and supplemental figures, more critical analysis of their data, and their response to the other reviewers' comments. I have no more outstanding comments.

Reviewer #3 (Remarks to the Author):

The authors addressed all my concerns, and the manuscript is now suitable for publication. The only comment I have is about point 6 (a bias in selection). I agree with the authors that a high correlation does not necessarily guarantee high decoding accuracy. However, it is certain that a no-correlation will lead to low decoding accuracy. Now, based on the criteria set by the authors: 1) there is a correlation between the BLA memory assembly and dCA1 ripple during post-training period (although high decoding accuracy is not guaranteed), but 2) there is no correlation in the period before training (hence, low decoding accuracy is guaranteed). If so, isn't the scale somehow tilted toward finding a difference between pre- and post-training? I would like the authors to elaborate on this and correct me if my assessment is inaccurate.

Reviewer #1 (Remarks to the Author):

The authors in their revision were able to address most of the concerns that I have raised. I think the revision has strengthened and improved the paper in several ways.

- 1) The inclusion of additional data in their analyses helped make the results more reliable.
- 2) The identification of the preconfiguration in the BLA assemblies that were later recruited with the hippocampal ripples in the post-sleep helped clarify the inter-regional consolidation mechanism.
- 3) The toning down of the claims regarding causation have strengthened the evidence based interpretation of the results. Speculation about the causation and proposal of new experiments to test the causation seem to be valid discussions based on their findings.

Except for the following three minor comments, I have no further comments:

- 1) Line 368; The authors state “score of 3.3 or greater for three or more consecutive bins (within ± 150 ms from ripple onset) were defined as ripple-modulated assemblies”, but rebuttal text says, “An assembly was considered ripple modulated if the correlogram peaks within 200 ms of ripple onset exceeded 3.3 s.d. above baseline ($P < 0.001$)”, please clarify the inconsistency. Also, please mention the criteria for selecting the BLA ripple modulated neurons (whether it is the same or different to the BLA ripple modulated assemblies).

We thank the reviewer for pointing out the mistake in our rebuttal letter. The correct time window should be ± 150 ms. Additionally, we confirm that the criteria for selecting ripple-modulated BLA neurons and ripple-modulated BLA assemblies are the same. We have clarified this in the revised manuscript.

- 2) In the rebuttal authors state, “For the same reasons discussed above, we now include statistical analyses for both “memory assemblies” and “memory neurons” (Fig. 5).” Fig 5 does not seem to have that information. I can find it in Fig 4 though. Please clarify.

In our revised manuscript, Fig .5 includes statistical analyses for both “memory assemblies” (Fig. 5c) and “memory neurons” (Fig. 5d).

- 3) Some of the key references are missing and should be added when discussing cell assembly detection and experience-dependent reactivation (Peyrache et al., 2009; Farooq et al. 2019) and network preconfiguration and neuronal selection (Dragoi and Tonegawa, 2011; Dragoi, 2024).

We thank the reviewer for the suggestion and have added these recommended citations into our revised manuscript.

Reviewer #2 (Remarks to the Author):

In the revised manuscript, the authors thoroughly addressed my previous comments. I was pleased by the expanded Discussion section, increased N for experimental animals, revised main and supplemental figures, more critical analysis of their data, and their response to the other reviewers' comments. I have no more outstanding comments.

Reviewer #3 (Remarks to the Author):

The authors addressed all my concerns, and the manuscript is now suitable for publication. The only comment I have is about point 6 (a bias in selection). I agree with the authors that a high correlation

does not necessarily guarantee high decoding accuracy. However, it is certain that a no-correlation will lead to low decoding accuracy. Now, based on the criteria set by the authors: 1) there is a correlation between the BLA memory assembly and dCA1 ripple during post-training period (although high decoding accuracy is not guaranteed), but 2) there is no correlation in the period before training (hence, low decoding accuracy is guaranteed). If so, isn't the scale somehow tilted toward finding a difference between pre- and post-training? I would like the authors to elaborate on this and correct me if my assessment is inaccurate.

We agree with the reviewer that the decoding accuracy likely favors the post-training period dataset. A potential follow-up experiment could involve inducing strong vs. weak memories by using varying intensities of footshocks, as suggested by Reviewer #2 in their original comments. In this approach, correlating decoding accuracies with memory strengths could provide further evidence supporting the role of dCA1-to-BLA information flow in memory formation.